# Linear-Core Surrogates: Smooth Loss Functions with Linear Rates for Classification and Structured Prediction

**Mehryar Mohri** [1] [2]  **Yutao Zhong** [1]

## Abstract

A fundamental dichotomy in the theory of classification sets smoothness against statistical efficiency: smooth surrogate losses such as the logistic loss enable fast $O(1/T)$ optimization but yield slow square-root $\mathcal{H}$-consistency bounds, while piecewise-linear losses like the Hinge loss achieve optimal linear $\mathcal{H}$-consistency rates but are non-differentiable. We introduce *Linear-Core (LC) Surrogates*, the first family of *explicit* convex loss functions that provably resolve this tension. By stitching a linear core to a smooth tail, we construct surrogates that are differentiable everywhere ($C^1$, and even $C^2$ under mild conditions) while retaining strict linear $\mathcal{H}$-consistency bounds, the strongest known form of consistency guarantee. We establish these linear bounds across three increasingly complex settings: binary classification, multi-class classification, and structured prediction. To our knowledge, this is the first explicit construction to simultaneously achieve smoothness and linear $\mathcal{H}$-consistency in any of these settings. Beyond their theoretical appeal, Linear-Core Surrogates offer practical advantages. In multi-class classification, their constant gradient profile near the decision boundary provides natural robustness to instance-dependent label noise, outperforming Cross-Entropy by *2.6%* on corrupted CIFAR-10. In structured prediction, their smoothness enables an unbiased stochastic gradient estimator that bypasses the $O(|\mathcal{Y}|^2)$ per-step complexity of exact inference, yielding a *23× speedup* over Structured SVMs on large-vocabulary sequence tagging tasks.

[1]Google Research, New York, NY; [2]Courant Institute of Mathematical Sciences, New York, NY. Correspondence to: Mehryar Mohri <mohri@google.com>, Yutao Zhong <yutaozhong@google.com>.

*Proceedings of the $43^{rd}$ International Conference on Machine Learning*, Seoul, South Korea. PMLR 306, 2026. Copyright 2026 by the author(s).

## 1. Introduction

In the theory of classification, the choice of loss function is governed by two conflicting desiderata: computational tractability and statistical consistency. On one hand, practical optimization requires loss functions that are convex and smooth, enabling the use of efficient gradient-based algorithms with fast convergence rates ($O(1/T)$ or better) (Nesterov, 1983; Beck & Teboulle, 2009). On the other hand, theoretical guarantees often rely on *consistency bounds*, which relate the excess error of the surrogate to that of the discrete target loss, e.g., the 0-1 loss. While classical *Bayes-consistency* ensures convergence in the infinite-sample limit and for the family of all measurable functions (Zhang, 2004; Lin, 2004; Bartlett et al., 2006; Steinwart, 2007), recent work has focused on the stronger notion of $\mathcal{H}$-*consistency*, which provides non-asymptotic guarantees restricted to the hypothesis set $\mathcal{H}$ of interest (Awasthi, Mao, Mohri, and Zhong, 2022a;b; Mao, Mohri, and Zhong, 2023e; 2024a).

Historically, a dichotomy has persisted in this landscape: smooth losses like the Logistic (Cross-Entropy) or Exponential loss yield valid consistency bounds, but these bounds are notoriously slow. Specifically, due to the vanishing curvature of these losses near the origin, the transfer rate is on the order of the square root of the excess surrogate error ($\Delta\mathcal{R} \propto \sqrt{\Delta\mathcal{L}}$) (Bartlett et al., 2006; Zhang, 2004). This implies that high precision in optimization translates inefficiently to target accuracy. Conversely, piecewise-linear losses like the Hinge loss offer fast linear consistency rates ($\Delta\mathcal{R} \propto \Delta\mathcal{L}$) (Steinwart, 2007; Awasthi et al., 2022a) but suffer from non-differentiability, leading to optimization instability and slower sub-gradient convergence rates ($O(1/\sqrt{T})$) (see Table 1 for a comparison).

In this work, we propose *Linear-Core (LC) Surrogates*, a new family of *explicit* smooth loss functions designed to resolve this dichotomy. By constructing a loss with a linear "core" stitched to a smooth tail, we achieve the best of both worlds: the fast $O(1/T)$ optimization rates of smooth losses and the optimal linear $\mathcal{H}$-consistency bounds of the Hinge loss. To our knowledge, this is the first explicit construction of smooth surrogates with provable linear $\mathcal{H}$-consistency bounds.

We extend the Linear-Core framework to the challenging domain of structured prediction, where the output space $\mathcal{Y}$ is exponentially large. Standard surrogates such as the Structured SVM (SSVM) loss (Tsochantaridis et al., 2005) or the CRF negative log-likelihood (Lafferty et al., 2001) often lack consistency guarantees with respect to discrete target metrics like the Hamming loss (Ciliberto et al., 2016; Osokin et al., 2017; Nowak et al., 2019). Mao et al. (2023e) addressed this theoretical gap by providing a detailed analysis of $\mathcal{H}$-consistency for structured prediction and introducing a family of structured losses with provable $\mathcal{H}$-consistency guarantees, though with square-root bounds. Building on this foundation, we show that our Linear-Core surrogates not only preserve these rigorous guarantees but *strengthen* them to *linear* $\mathcal{H}$-consistency bounds. As an additional practical benefit, their smoothness enables an unbiased stochastic gradient estimator that bypasses the $O(|\mathcal{Y}|^2)$ bottleneck of exact inference in standard SSVM and CRF methods.

Our approach is most closely related to the work of Cao et al. (2025), which also seeks linear convergence rates via *Convolutional Fenchel-Young losses*. While their resulting binary loss profile structurally resembles our construction, their general framework defines losses *implicitly* via variational optimization, relies on non-standard decoding, and is limited to excess loss bounds with respect to the family of all measurable functions (Bayes-consistency). In contrast, our Linear-Core surrogates are *explicit*, standard convex functions compatible with standard $\mathrm{argmax}$ decoding, and we provide the strictly stronger $\mathcal{H}$-*consistency* bounds. Furthermore, our framework extends to multi-class and structured prediction with closed-form losses, and the explicit formulation enables our unbiased stochastic sampling algorithm (Section 5.3), which bypasses the $O(|\mathcal{Y}|^2)$ bottleneck of exact inference. See Appendix A for an extended discussion.

The remainder of this paper follows a deliberate progression of the label space complexity, where each setting builds upon the previous one. Our primary contribution is theoretical: we resolve a fundamental open question by constructing the first explicit smooth surrogates with linear $\mathcal{H}$-consistency, and we do so across all three settings. *Binary Classification (Section 3):* We establish the fundamental mechanism in the simplest setting ($|\mathcal{Y}| = 2$), constructing the Linear-Core surrogate by stitching a linear core to a smooth tail and proving that it achieves both $C^1/C^2$ smoothness and linear $\mathcal{H}$-consistency bounds (Sections 3.2). To our knowledge, this is the first explicit smooth surrogate with provable linear $\mathcal{H}$-consistency. *Multi-Class Classification (Section 4):* We scale to $n$ competing classes via sum-losses, proving that the multi-class objective inherits the optimal linear transfer bound (Section 4.2). The constant gradient profile near the decision boundary acts as a natural 'hard' regularizer, yielding +2.6% improvement over Cross-Entropy on CIFAR-10 under instance-dependent

*Table 1.* Comparison of surrogate loss properties.

| Loss Function | Convexity | Smoothness | Consistency Bound | Rate |
|---|---|---|---|---|
| Hinge Loss | Yes | No (Non-diff. at $\pm 1$) | Linear | Fast |
| Squared-Hinge | Yes | $C^1$ | Square-root | Slow |
| Logistic / Exp | Yes | $C^\infty$ | Square-root | Slow |
| **Linear-Core ($\bar{\Phi}$)** | **Yes** | $C^1/C^2$ | **Linear** | **Fast** |

noise (Section 4.3). *Structured Prediction (Section 5):* We tackle the most complex domain where the output space is exponentially large, proving linear $\mathcal{H}$-consistency bounds for structured sum-losses (Section 5.2). The smoothness of our surrogates enables an unbiased stochastic gradient estimator with bounded variance independent of $|\mathcal{Y}|$ (Theorem 5.3), which bypasses the $O(|\mathcal{Y}|^2)$ bottleneck of exact inference, achieving a 23× speedup over SSVM and 17.4× over CRF (Sections 5.4 and 5.5).

## 2. Preliminaries

We denote the input space by $\mathcal{X}$, the label space by $\mathcal{Y}$, the hypothesis set by $\mathcal{H}$ and the data distribution by $\mathcal{D}$. We consider a *target loss* $\ell_{\mathrm{tar}}: \mathcal{H} \times \mathcal{X} \times \mathcal{Y} \rightarrow \mathbb{R}$ (e.g., the 0-1 loss) and a *surrogate loss* $\ell_{\mathrm{sur}}: \mathcal{H} \times \mathcal{X} \times \mathcal{Y} \rightarrow \mathbb{R}$ (e.g., a convex margin loss). The *generalization error* of a hypothesis $h \in \mathcal{H}$ is defined as $\mathcal{E}_\ell(h) = \mathbb{E}_{(x,y)\sim\mathcal{D}}[\ell(h,x,y)]$. The *best-in-class error* is denoted by $\mathcal{E}_\ell^*(\mathcal{H}) = \inf_{h\in\mathcal{H}} \mathcal{E}_\ell(h)$. The difference $\mathcal{E}_\ell(h) - \mathcal{E}_\ell^*(\mathcal{H})$ is referred to as the *estimation error*. We analyze $\mathcal{H}$-*consistency bounds* (Awasthi et al., 2022a; Mao et al., 2023f), which relate the estimation error of the target loss to that of the surrogate. Such bounds typically take the form: $\mathcal{E}_{\ell_{\mathrm{tar}}}(h) - \mathcal{E}_{\ell_{\mathrm{tar}}}^*(\mathcal{H}) + \mathcal{M}_{\ell_{\mathrm{tar}}}(\mathcal{H}) \leq \Gamma\left(\mathcal{E}_{\ell_{\mathrm{sur}}}(h) - \mathcal{E}_{\ell_{\mathrm{sur}}}^*(\mathcal{H}) + \mathcal{M}_{\ell_{\mathrm{sur}}}(\mathcal{H})\right)$, where $\Gamma$ with $\Gamma(0) = 0$ is an increasing concave function (e.g., $t \mapsto t$ or $t \mapsto \sqrt{t}$). The term $\mathcal{M}_\ell(\mathcal{H})$ is the *minimizability gap*, defined as $\mathcal{M}_\ell(\mathcal{H}) = \mathcal{E}_\ell^*(\mathcal{H}) - \mathbb{E}_x[\inf_{h\in\mathcal{H}} \mathbb{E}_y[\ell(h,x,y) \mid x]]$. This quantity measures the discrepancy between the best possible expected loss within $\mathcal{H}$ and the expected pointwise infimum. It is upper bounded by, yet generally finer than, the standard approximation error $\mathcal{E}_\ell^*(\mathcal{H}) - \mathcal{E}_\ell^*(\mathcal{H}_{\mathrm{all}})$, where $\mathcal{H}_{\mathrm{all}}$ denotes the family of all measurable functions (Mao et al., 2024a). We denote the excess target and surrogate errors by $\Delta\mathcal{R} = \mathcal{E}_{\ell_{\mathrm{tar}}}(h) - \mathcal{E}_{\ell_{\mathrm{tar}}}^*(\mathcal{H}_{\mathrm{all}})$ and $\Delta\mathcal{L} = \mathcal{E}_{\ell_{\mathrm{sur}}}(h) - \mathcal{E}_{\ell_{\mathrm{sur}}}^*(\mathcal{H}_{\mathrm{all}})$, respectively. When $\mathcal{H} = \mathcal{H}_{\mathrm{all}}$, the minimizability gap reduces to the approximation error. Consequently, in this case, an $\mathcal{H}$-consistency bound implies the standard excess error bound $\Delta\mathcal{R} \leq \Gamma(\Delta\mathcal{L})$. Thus, $\mathcal{H}$-consistency bound is a strictly stronger guarantee than standard Bayes-consistency.

## 3. Binary Classification

We first consider the binary classification setting where $\ell_{\mathrm{tar}}$ is the binary zero-one loss, defined by $\ell_{0-1}(h,x,y) = 1_{\mathrm{sign}(h(x))\neq y}$. When the target is the binary zero-one loss, we define the $\mathcal{H}$-*estimation error transformation* $\mathcal{T}$ as the function satisfying the following tight lower bound

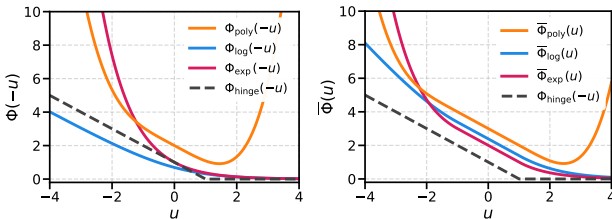

*Figure 1.* Left: Standard surrogates. Right: LC surrogates $\overline{\Phi}$.

for all $h \in \mathcal{H}$: $\mathcal{T}\big(\mathcal{E}_{\ell_{0-1}}(h) - \mathcal{E}^*_{\ell_{0-1}}(\mathcal{H}) + \mathcal{M}_{\ell_{0-1}}(\mathcal{H})\big) \leq \mathcal{E}_{\ell_{\mathrm{sur}}}(h) - \mathcal{E}^*_{\ell_{\mathrm{sur}}}(\mathcal{H}) + \mathcal{M}_{\ell_{\mathrm{sur}}}(\mathcal{H})$. Tightness implies that for any $t \in [0,1]$, there exists a distribution and a hypothesis such that $\mathcal{E}_{\ell_{0-1}}(h) - \mathcal{E}^*_{\ell_{0-1}}(\mathcal{H}) + \mathcal{M}_{\ell_{0-1}}(\mathcal{H}) = t$ and $\mathcal{E}_{\ell_{\mathrm{sur}}}(h) - \mathcal{E}^*_{\ell_{\mathrm{sur}}}(\mathcal{H}) + \mathcal{M}_{\ell_{\mathrm{sur}}}(\mathcal{H}) = \mathcal{T}(t)$. Explicit forms of $\mathcal{T}$ have been characterized for binary margin-based losses (Awasthi et al., 2022a; Mao et al., 2024a).

### 3.1. Smooth Surrogates with Linear Cores

Let $\Phi \colon \mathbb{R} \to \mathbb{R}_+$ be a differentiable convex function with $\Phi'(0) > 0$. For instance, one may take $\Phi(u) = \log(1 + e^u)$, corresponding to the logistic loss, or $\Phi(u) = e^u$, corresponding to the exponential loss. We define the smooth *linear-core (LC) surrogate* loss as the margin-based loss function $\ell_{\overline{\Phi}} \colon (h, x, y) \mapsto \overline{\Phi}(yh(x))$, where $\overline{\Phi}$ is given by

$$\overline{\Phi}(u) = \begin{cases} -u + 1 + \dfrac{\Phi(0)}{\Phi'(0)}, & -1 \leq u \leq 1, \\ \dfrac{\Phi(1-u)}{\Phi'(0)}, & u > 1, \\ \dfrac{\Phi(-1-u)}{\Phi'(0)} + 2, & u < -1. \end{cases} \tag{1}$$

Figure 1 gives several illustrations of $\overline{\Phi}$. We also consider a *one-sided* linear-core surrogate, obtained by smoothing only the right outer branch and keeping the left side linear. We define $\widetilde{\Phi}$ as:

$$\widetilde{\Phi}(u) = \begin{cases} -u + 1 + \dfrac{\Phi(0)}{\Phi'(0)}, & u \leq 1, \\ \dfrac{\Phi(1-u)}{\Phi'(0)}, & u > 1. \end{cases} \tag{2}$$

Note that any constant can be added to the loss function without affecting the minimization. The following proposition establishes the key analytical properties for the entire family of Linear-Core surrogates (See Appendix B for proofs).

**Proposition 3.1** (Convexity and Smoothness). *Let $\phi \in \{\overline{\Phi}, \widetilde{\Phi}\}$. The Linear-Core surrogates (Binary $\phi$, Multi-class $\ell^{\mathrm{sum}}_\phi$, and Structured $\mathsf{L}^{\mathrm{sum}}_\phi$ defined in Sections 4 and 5) are globally convex and continuously differentiable ($C^1$). Furthermore, if the base $\Phi$ satisfies $\Phi''(0) = 0$, they are twice continuously differentiable ($C^2$).*

The condition $\Phi''(0) = 0$ is not vacuous. For example, take $\Phi(u) = a\,u + \frac{1}{12}u^4 + K$ with $a > 0$. Then $\Phi''(0) = 0$, and

by Proposition 3.1, the corresponding linear-core surrogate is twice continuously differentiable on $\mathbb{R}$. In contrast, for common choices such as the logistic loss $\Phi(u) = \log(1+e^u)$ or the exponential loss $\Phi(u) = e^u$, one has $\Phi''(0) > 0$, so $\overline{\Phi}$ is $C^1$ but not $C^2$ at the hinge points $u = \pm 1$.

### 3.2. Linear $\mathcal{H}$-Consistency Bound

We call a hypothesis set $\mathcal{H}$ complete if, for every $x \in \mathcal{X}$, $\{h(x) \colon h \in \mathcal{H}\} = \mathbb{R}$. Since $\overline{\Phi}$ is convex and differentiable at zero and satisfies the inequality $\overline{\Phi}'(0) = -1 < 0$, by Mao et al. (2024a, Theorem 4.1), for complete hypothesis sets, the tight transformation $\mathcal{T}$ takes the following form.

**Theorem 3.2.** *Let $\mathcal{H}$ be a complete hypothesis set. The transformation $\mathcal{T}$ can be expressed as follows:*

$$\forall t \in [0,1], \quad \mathcal{T}(t) = \overline{\Phi}(0) - \inf_{u \in \mathbb{R}}\big( \tfrac{1-t}{2}\overline{\Phi}(-u) + \tfrac{1+t}{2}\overline{\Phi}(u)\big).$$

See Appendix G.1 for a proof. The following result shows that the transformation is bounded below by a linear function of $t$. The proof is presented in Appendix G.2.

**Lemma 3.3.** *For all $t \in [0,1]$, $\mathcal{T}(t) \geq t$ and $\mathcal{T}(0) = 0$.*

By Lemma 3.3 together with Awasthi et al. (2022a, Theorem 4), we obtain a linear $\mathcal{H}$-consistency bound for the surrogate losses $\ell_{\overline{\Phi}}$.

**Theorem 3.4** (Linear $\mathcal{H}$-consistency bound). *Let $\mathcal{H}$ be a complete hypothesis set. Then, for all $h \in \mathcal{H}$,*

$$\begin{aligned} \mathcal{E}_{\ell_{0-1}}(h) &- \mathcal{E}^*_{\ell_{0-1}}(\mathcal{H}) + \mathcal{M}_{\ell_{0-1}}(\mathcal{H}) \\ &\leq \mathcal{E}_{\ell_{\overline{\Phi}}}(h) - \mathcal{E}^*_{\ell_{\overline{\Phi}}}(\mathcal{H}) + \mathcal{M}_{\ell_{\overline{\Phi}}}(\mathcal{H}). \end{aligned}$$

*Proof Sketch.* The proof relies on the $\mathcal{H}$-consistency framework of Awasthi et al. (2022a). The key step is analyzing the transformation function $\mathcal{T}(t)$, which relates the surrogate estimation error to the target error. Since our surrogate $\overline{\Phi}$ is linear with slope $-1$ on the interval $[-1, 1]$, we show that the $\mathcal{T}$ satisfies the lower bound $\mathcal{T}(t) \geq t$ for all $t \in [0, 1]$ (Lemma 3.3). This linear lower bound directly implies the linear consistency rate $\Delta\mathcal{R} \leq \Delta\mathcal{L}$ when $\mathcal{H} = \mathcal{H}_{\mathrm{all}}$. $\qquad\square$

Intuition: Unlike smooth losses where the gradient vanishes at the origin (causing "flat" landscapes and slow $\sqrt{t}$ transfer), the Linear-Core surrogate maintains a non-zero gradient near the decision boundary. This forces the surrogate estimation error to scale linearly with the target error.

Combining Proposition 3.1 with Theorem 3.4, we obtain convex and smooth (even twice continuously differentiable) surrogate losses with linear $\mathcal{H}$-consistency bounds. Note that this does not conflict with Mao et al. (2024a, Theorem 4.2), since $\overline{\Phi}''(0) = 0$.

## 3.3. One-sided smoothing

We next analyze the consistency of the one-sided linear-core surrogate $\widetilde{\Phi}$ defined in Eq. (2). By Proposition 3.1, $\widetilde{\Phi}$ is convex and smooth. The linear lower bound on the transformation also holds for this variant.

**Lemma 3.5** (Linear bound for one-sided smoothing). *For $t \in [0,1]$, define*

$$\mathcal{T}_{\text{one}}(t) = \widetilde{\Phi}(0) - \inf_{u \in \mathbb{R}} \Big( \tfrac{1-t}{2} \, \widetilde{\Phi}(-u) + \tfrac{1+t}{2} \, \widetilde{\Phi}(u) \Big).$$

*Then $\mathcal{T}_{\text{one}}(t) \geq t$ and $\mathcal{T}_{\text{one}}(0) = 0$.*

The proof, given in Appendix G.4, is essentially identical to the proof of Lemma 3.3, since both arguments rely only on the linear core of $\widetilde{\Phi}$ over $[-1,1]$. This shows that the fundamental linear lower bound carries over unchanged to the one-sided case.

Furthermore, Theorem 3.4 also remains valid without modification, since its proof relies only on Lemma 3.3, which we have extended to the one-sided smoothing case in Lemma 3.5. The proof is presented in Appendix G.5.

**Corollary 3.6** (Linear $\mathcal{H}$-consistency bound for one-sided smoothing). *Let $\mathcal{H}$ be a complete hypothesis set. Then for $\widetilde{\Phi}$, the following linear $\mathcal{H}$-consistency bound holds:*

$$\forall h \in \mathcal{H}, \quad \mathcal{E}_{\ell_{0-1}}(h) - \mathcal{E}^*_{\ell_{0-1}}(\mathcal{H}) + \mathcal{M}_{\ell_{0-1}}(\mathcal{H})$$
$$\leq \mathcal{E}_{\ell_{\widetilde{\Phi}}}(h) - \mathcal{E}^*_{\ell_{\widetilde{\Phi}}}(\mathcal{H}) + \mathcal{M}_{\ell_{\widetilde{\Phi}}}(\mathcal{H}).$$

Intuitively, $\widetilde{\Phi}$ inherits all desirable properties of $\overline{\Phi}$ (convexity, smoothness, and linear $\mathcal{H}$-consistency bounds) while smoothing only one side. This is attractive when one prefers to soften the penalty for large positive margins while keeping the negative side linear, e.g., when the objective tolerates large positive scores but requires sharper control on the negative side.

## 3.4. Empirical Validation of Convergence Rates

To illustrate the theoretical distinction between the linear $\mathcal{H}$-consistency of our Linear-Core Surrogates and the slower consistency of standard smooth losses, we analyze a canonical *biased coin* problem (Bartlett et al., 2006). We consider a binary classification task where the label probability is $\eta = 1/2 + \delta$, with $\delta > 0$ representing the margin.

We compute the exact excess surrogate error $\Delta\mathcal{L}$ and excess target error $\Delta\mathcal{R}$ analytically across a range of margins $\delta \in [10^{-4}, 10^{-1}]$. This setup removes finite-sample optimization noise and isolates the asymptotic convergence behavior of the loss functions. Figure 2 reports the results.

The log-log plot confirms that the Linear-Core Surrogate (blue) maintains a strict linear relationship (slope $\approx$ 1) where $\Delta\mathcal{R} = O(\Delta\mathcal{L})$. In contrast, the standard Logistic loss (red) exhibits the slower square-root relationship (slope $\approx$

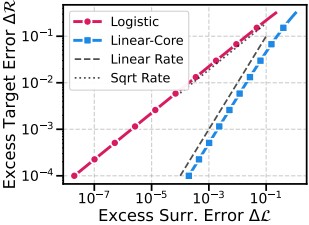

*Figure 2.* Rates: LC vs. Logistic.

0.5) characteristic of losses with vanishing curvature, where $\Delta\mathcal{R} = O(\sqrt{\Delta\mathcal{L}})$. This confirms that for hard problems (small $\delta$), minimizing the Linear-Core Surrogate translates to target error reduction significantly faster than minimizing the logistic loss. Appendix C further considers a generalized family parameterized by $\tau > 0$, showing that the linear rates are robust to this choice.

## 4. Multi-Class Classification

Let $\mathcal{Y}$ be a finite label set with cardinality $|\mathcal{Y}| = n \geq 2$. We consider score functions $h: \mathcal{X} \times \mathcal{Y} \to \mathbb{R}$, where the vector $h(x, \cdot) \in \mathbb{R}^n$ represents the scores assigned to each class. For any instance $(x, y) \in \mathcal{X} \times \mathcal{Y}$, we define the pairwise margins as: $m_{y,y'}(x) := h(x, y) - h(x, y')$ for all $y' \in \mathcal{Y}$. The target $\ell_{\text{tar}}$ is typically the multi-class zero-one loss $\mathsf{L}_{0-1}$, defined as $\mathsf{L}_{0-1}(h, x, y) = 1_{\mathsf{h}(x) \neq y}$, where $\mathsf{h}(x) = \operatorname{argmax}_{y' \in \mathcal{Y}} h(x, y')$ denotes the label predicted by $h$ for the input $x$.

**Sum losses.** Generalizing the formulation of (Weston & Watkins, 1998), we work with *sum* surrogates of the form

$$\ell_{\Phi}^{\text{sum}}(h, x, y) = \sum_{y' \neq y} \Phi(-m_{y,y'}(x)). \tag{3}$$

(Equivalently, one may sum over $y' \in \mathcal{Y}$; this choice differs only by an additive constant when $\Phi(0)$ is finite and does not affect minimization.) We define the multi-class smooth surrogates by replacing $\Phi$ in (3) with either the symmetric linear-core surrogate $\overline{\Phi}$ or the one-sided smoothing $\widetilde{\Phi}$:

$$\ell_{\overline{\Phi}}^{\text{sum}}(h, x, y) = \sum_{y' \neq y} \overline{\Phi}(m_{y,y'}(x)),$$
$$\ell_{\widetilde{\Phi}}^{\text{sum}}(h, x, y) = \sum_{y' \neq y} \widetilde{\Phi}(m_{y,y'}(x)).$$

## 4.1. Convexity and smoothness.

As established in Proposition 3.1, these multi-class surrogates preserve the desirable analytical properties of the original binary surrogates. The losses $\ell_{\overline{\Phi}}^{\text{sum}}$ and $\ell_{\widetilde{\Phi}}^{\text{sum}}$ are convex in the score vector $h(x, \cdot)$ and are globally $C^1$, and even $C^2$ under mild conditions on $\Phi$. Figure 3 visualizes these surfaces for a three-class problem, plotting the loss as a function of the pairwise margins $m_1$ and $m_2$; the linear plateau in the central region reflects the linear-core structure. Explicit closed-form expressions are given in Appendix D.

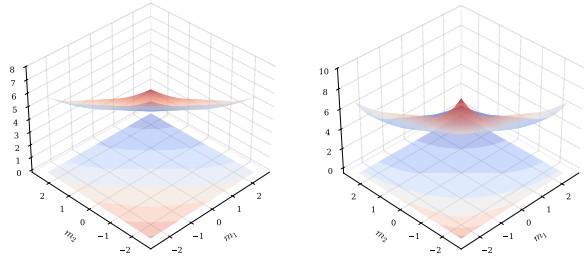

*Figure 3.* Multi-class sum loss surfaces $\ell_\Phi^{\text{sum}}$ for a 3-class problem. Left: Logistic LC Surrogate. Right: Exponential LC Surrogate.

## 4.2. Linear $\mathcal{H}$-Consistency Bound

We now establish a linear $\mathcal{H}$-consistency bound with respect to the 0-1 loss. In contrast with the squared-hinge analysis (which leads to a $\sqrt{\cdot}$ bound), the linear-core surrogates yield an exact linear bound because their central branch is affine, guaranteeing fast-rate convergence to the best-in-class classifier under minimal assumptions.

**Theorem 4.1** (Linear $\mathcal{H}$-consistency bound for multi-class linear-core surrogates). *Assume $\mathcal{H}$ is symmetric and complete. Then, for any distribution and any $h \in \mathcal{H}$,*

$$\mathcal{R}_{\mathsf{L}_{0-1}}(h) - \mathcal{R}_{\mathsf{L}_{0-1}}^*(\mathcal{H}) + \mathcal{M}_{\mathsf{L}_{0-1}}(\mathcal{H})$$
$$\leq \mathcal{R}_{\ell_{\overline{\Phi}}^{\text{sum}}}(h) - \mathcal{R}_{\ell_{\overline{\Phi}}^{\text{sum}}}^*(\mathcal{H}) + \mathcal{M}_{\ell_{\overline{\Phi}}^{\text{sum}}}(\mathcal{H}),$$
$$\mathcal{R}_{\mathsf{L}_{0-1}}(h) - \mathcal{R}_{\mathsf{L}_{0-1}}^*(\mathcal{H}) + \mathcal{M}_{\mathsf{L}_{0-1}}(\mathcal{H})$$
$$\leq \mathcal{R}_{\ell_{\widetilde{\Phi}}^{\text{sum}}}(h) - \mathcal{R}_{\ell_{\widetilde{\Phi}}^{\text{sum}}}^*(\mathcal{H}) + \mathcal{M}_{\ell_{\widetilde{\Phi}}^{\text{sum}}}(\mathcal{H}).$$

*Proof Sketch.* We decompose the conditional surrogate regret (see Appendix F for the exact definition) into a sum of pairwise regrets between labels. By lower bounding the total regret using only the specific pair $(y_{\max}, \mathsf{h}(x))$, we reduce the problem to the binary case. Since the Linear-Core loss has a constant gradient of magnitude 1 at the origin (unlike the vanishing gradient of smooth losses), the pairwise regret provides a linear lower bound on the probability difference $p(y_{\max} \mid x) - p(\mathsf{h}(x) \mid x)$, which corresponds exactly to the 0-1 conditional regret (Lemma H.1 in Appendix H.1). $\square$

The *sum* formulation in (3) aggregates pairwise margins against all competing labels, which has two pleasant consequences in our setting. First, replacing $\Phi$ by either linear-core surrogate $\overline{\Phi}$ or $\widetilde{\Phi}$ preserves convexity in the score vector $h(x, \cdot)$ and grants global $C^1$-smoothness, and even $C^2$-smoothness under mild assumptions on $\Phi$ (Proposition 3.1). Second, the pairwise structure lets us reduce conditional regret lower bounds to a family of *two-class* one-dimensional optimization problems that can be solved in closed form (Lemmas H.2 in Appendix H.1). Compared with sum squared-hinge/exponential surrogates (which yield a $\sqrt{\cdot}$ transfer) (Awasthi et al., 2022b), the linear-core surrogates admit a *linear $\mathcal{H}$-consistency bound* because their

middle branch is affine with slope $-1$ at the origin; this ensures that the pointwise supporting-line lower bound holds.

## 4.3. Empirical Validation: Robustness to Noise

While Section 4 established the theoretical properties of our Linear-Core surrogates, their practical use is best demonstrated by their robustness to realistic data corruption. Standard losses like Cross-Entropy (CE), also known as logistic loss (Verhulst, 1838; 1845; Berkson, 1944; 1951), are particularly sensitive to *Instance-Dependent Noise (IDN)* (Berthon et al., 2021; Cheng et al., 2020; Du & Cai, 2015). Unlike the uniform noise model (also known as symmetric label noise) (van Rooyen et al., 2015; Ghosh et al., 2017), IDN concentrates corruption near the decision boundary, where the probability of mislabeling correlates with feature ambiguity (e.g., an image of a "Dog" resembling a "Wolf"). This reflects a far more realistic noise model encountered in real-world applications, as human annotators rarely mislabel unambiguous examples far from the decision boundary (Xia et al., 2020).

We hypothesize that the robustness of the One-Sided Linear-Core surrogate stems from its gradient saturation near this critical boundary region. As illustrated in Figure 4, the Cross-Entropy loss (logistic loss) has non-zero curvature ($\Phi'' > 0$) at the margin $u = 0$, and its gradient magnitude varies continuously with the distance to the boundary.

This allows the optimizer to reduce the total loss by making fine-grained shifts to the decision boundary to accommodate ambiguous, noisy examples. In contrast, the One-Sided Linear-Core surrogate is strictly affine for all margins $u \leq 1$. Consequently, its gradient is locally constant (invariant) with respect to the margin for misclassified and near-boundary examples (see Figure 4, blue line). This prevents the optimizer from shifting the decision boundary to minimize the individual losses of corrupted points, effectively acting as a 'hard' regularizer that counts margin violations rather than fitting their probability estimates.

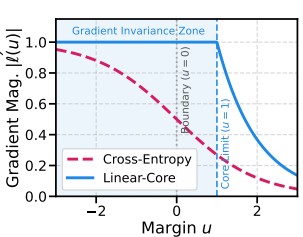

*Figure 4.* Gradient magnitudes: Linear-Core (Blue, saturated) vs. Logistic (Red, variable).

To validate this, we compare our surrogate against the Cross-Entropy baseline on CIFAR-10 (Krizhevsky, 2009) under feature-dependent label noise. We also compare our method against the *Generalized Cross-Entropy (GCE)* loss (Zhang & Sabuncu, 2018), which has been shown to be empirically robust against uniform label noise.

**Setup.** We train a ResNet-18 (He et al., 2016) using Stochastic Gradient Descent (SGD) with momentum 0.9 (Nesterov,

*Table 2.* Test Accuracy on CIFAR-10 under Instance-Dependent Noise. Linear-Core outperforms baselines.

| IDN Rate ($\rho$) | Cross-Entropy (CE) | Gen. Cross-Entropy (GCE, Best Tuned) | Linear-Core (Ours) | Improvement (vs. GCE) |
|---|---|---|---|---|
| 20% | 83.20 | 83.23 | **84.24** | +1.01 |
| 30% | 77.83 | 77.90 | **80.52** | +2.62 |
| 40% | 72.88 | 72.93 | **75.49** | +2.56 |
| 50% | 61.08 | 61.14 | **63.57** | +2.43 |
| 60% | 37.50 | 37.57 | **39.86** | +2.29 |

1983), weight decay $5 \times 10^{-4}$, and a batch size of 128. The learning rate is initialized at $0.1$ and annealed using a cosine schedule for 50 epochs. We introduce instance-dependent noise following the protocol of Xia et al. (2020): we project image features onto a random decision boundary to generate flip probabilities, ensuring that visually ambiguous images are significantly more likely to be mislabeled. We test noise rates $\rho \in \{20\%, 30\%, 40\%, 50\%, 60\%\}$. We perform a grid search for the hyperparameter $q \in (0, 1]$ of the GCE loss and report the best performance for each noise rate to ensure a strong baseline.

**Results.** Table 2 summarizes the results. The weakness of GCE under instance-dependent noise is evident: its performance tracks the standard CE baseline almost identically across all noise rates (e.g., a negligible $0.05\%$ difference at $40\%$ noise), confirming that simply re-weighting the loss is insufficient when noise mimics hard examples. Our Linear-Core surrogate, however, establishes a distinct performance gap. It consistently outperforms GCE and CE across the entire spectrum of noise rates. At a low noise rate ($20\%$), our method already demonstrates superior generalization with a $+1.01\%$ gain. The advantage becomes most pronounced at moderate noise levels ($30\%$–$40\%$), where our surrogate surpasses the tuned GCE baseline by approximately $2.6\%$. Crucially, this robustness is sustained even under severe corruption: at $50\%$ and $60\%$ noise, where the signal is heavily degraded, our method maintains a consistent lead of approximately $2.3\%$–$2.4\%$. These results confirm that the constant gradient in the linear core effectively suppresses the signal from systematically corrupted, near-boundary examples where GCE and CE fail.

### 4.3.1. MECHANISM ANALYSIS: GRADIENT INVARIANCE

To understand the source of this robustness, we analyzed the gradient dynamics of the loss functions during training. At epoch 40 (after the learning rate decay), we recorded the gradient magnitude $|\ell'(h(x))|$ for two distinct groups of training examples: *clean samples* (where the label is correct) and *noisy samples* on CIFAR-10 with 40% instance-dependent noise.

Figure 5 visualizes the distribution of these gradients, revealing a striking difference in behavior. As shown in Figure 5 (Left), the Cross-Entropy loss assigns a broad range of high-magnitude gradients (0.6 to 1.0) to noisy samples. This indicates that the loss function is actively "negotiating" with

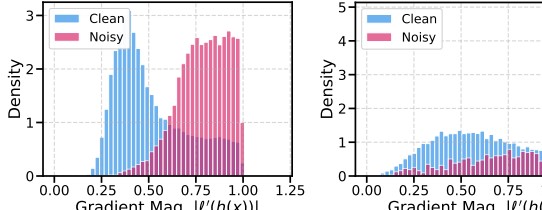

*Figure 5.* Gradient Magnitudes: CE (Left) vs. Linear-Core (Right). outliers, assigning variable penalties based on the model's confidence, which drives the decision boundary to overfit these corrupted points.

In contrast, Figure 5 (Right) shows that the Linear-Core surrogate exhibits a sharp, Dirac-like peak exactly at $|\ell'| \approx 1.0$ for noisy samples. This confirms our theoretical hypothesis: for samples with negative margins ($u \leq 1$), the gradient saturates and becomes *invariant* to the magnitude of the error. This "hard" clipping effectively ignores the degree of "wrongness" for outliers, preventing the optimizer from shifting the boundary to accommodate mislabeled examples.

## 5. Structured Prediction

We consider the general structured prediction setting where the output space $\mathcal{Y}$ may be exponentially large. We consider a target loss $\mathsf{L}(h, x, y) = \ell(\mathsf{h}(x), y)$, where $\ell: \mathcal{Y} \times \mathcal{Y} \to \mathbb{R}_+$ is a non-negative auxiliary loss function (e.g., Hamming loss) such that $\ell(y, y) = 0$ for all $y \in \mathcal{Y}$.

Following Mao et al. (2023e), we work with *structured sum losses*. Let $\bar{\ell}(y', y) = 1 - \ell(y', y)$ denote the similarity score. The structured sum loss is defined as: $\forall (x, y) \in \mathcal{X} \times \mathcal{Y}$,

$$\mathsf{L}_{\Phi}^{\text{sum}}(h, x, y) = \sum_{y' \in \mathcal{Y}} \bar{\ell}(y', y) \sum_{y'' \neq y'} \Phi(h(x, y'') - h(x, y')).$$

This formulation effectively aggregates pairwise margins, weighted by the structural similarity between the candidate $y'$ and the true label $y$. We define the structured prediction smooth surrogates by replacing the base function $\Phi$ with either the symmetric linear-core surrogate $\overline{\Phi}$ or the one-sided smoothing $\widetilde{\Phi}$:

$$\mathsf{L}_{\overline{\Phi}}^{\text{sum}}(h, x, y) = \sum_{y' \in \mathcal{Y}} \bar{\ell}(y', y) \sum_{y'' \neq y'} \overline{\Phi}(m_{y', y''}(x)),$$

$$\mathsf{L}_{\widetilde{\Phi}}^{\text{sum}}(h, x, y) = \sum_{y' \in \mathcal{Y}} \bar{\ell}(y', y) \sum_{y'' \neq y'} \widetilde{\Phi}(m_{y', y''}(x)).$$

### 5.1. Convexity and Smoothness

The optimization landscape of the structured surrogate is determined by the properties of $\overline{\Phi}$ and $\widetilde{\Phi}$. Proposition 3.1 confirms that the structural aggregation preserves the convexity of the base scalar function, ensuring that the learning objective remains amenable to global minimization. Furthermore, it establishes that our structured surrogates are globally $C^1$, and $C^2$ under mild conditions on $\Phi$.

## 5.2. Linear $\mathcal{H}$-Consistency Bound

We now state the main consistency result for structured prediction. Similar to the multi-class setting, the affine behavior of the linear-core surrogates near the origin allows us to derive a linear bound relating the estimation error of the surrogate to that of the target structured loss.

**Theorem 5.1** (Linear $\mathcal{H}$-consistency bound for structured prediction). *Assume $\mathcal{H}$ is symmetric and complete. Let* $\mathsf{L}$ *be the target structured loss defined by* $\ell$. *Then, for any distribution and any $h \in \mathcal{H}$, the following bounds hold:*

$$\mathcal{R}_{\mathsf{L}}(h) - \mathcal{R}_{\mathsf{L}}^*(\mathcal{H}) + \mathcal{M}_{\mathsf{L}}(\mathcal{H})$$
$$\leq \mathcal{R}_{\mathsf{L}_{\overline{\Phi}}^{\text{sum}}}(h) - \mathcal{R}_{\mathsf{L}_{\overline{\Phi}}^{\text{sum}}}^*(\mathcal{H}) + \mathcal{M}_{\mathsf{L}_{\overline{\Phi}}^{\text{sum}}}(\mathcal{H}),$$
$$\mathcal{R}_{\mathsf{L}}(h) - \mathcal{R}_{\mathsf{L}}^*(\mathcal{H}) + \mathcal{M}_{\mathsf{L}}(\mathcal{H})$$
$$\leq \mathcal{R}_{\mathsf{L}_{\overline{\Phi}}^{\text{sum}}}(h) - \mathcal{R}_{\mathsf{L}_{\overline{\Phi}}^{\text{sum}}}^*(\mathcal{H}) + \mathcal{M}_{\mathsf{L}_{\overline{\Phi}}^{\text{sum}}}(\mathcal{H}).$$

*Proof Sketch.* The structured loss is defined as a sum of pairwise margins weighted by the structural distance $\overline{\ell}(y, y')$. We define a conditional regret that aggregates these pairwise terms. Similar to the multi-class case, we lower bound this sum by the contribution of the "most violated" pair relative to the prediction. Because the local margin loss is linear near zero, this contribution scales linearly with the target structural error, avoiding the square-root degradation typical of sums of smooth convex functions. $\square$

In contrast to structured sum-exponential surrogates, which yield a square-root rate (Mao et al., 2023e), this result establishes a *linear* rate. The structured linear-core surrogates thus serve as valid smooth proxies for the discrete structured error with strictly stronger $\mathcal{H}$-consistency guarantees.

*Remark* 5.2 (Comparison of Convergence Rates). To explicitly compare our theorems to previous works, we highlight the theoretical difference in the convergence transfer rate dictated by the geometry of the surrogate losses:

- **Binary Classification:** Previous works analyzing standard smooth surrogates (e.g., Logistic, Exponential, Squared-Hinge) establish a square-root $\mathcal{H}$-consistency bound, meaning the excess target risk $\Delta\mathcal{R}$ is bounded by $O(\sqrt{\Delta\mathcal{L}})$ (Bartlett et al., 2006; Awasthi et al., 2022a). Piecewise-linear losses like the Hinge loss achieve a fast linear bound ($\Delta\mathcal{R} \leq O(\Delta\mathcal{L})$) but are non-differentiable (Awasthi et al., 2022a). Our Theorem 3.4 proves that the Linear-Core surrogate achieves the strict linear rate of the Hinge loss while remaining globally $C^1/C^2$ smooth.

- **Multi-Class Classification:** For multi-class sum losses, previous smooth variants (e.g., sum-exponential or sum-squared-hinge) similarly suffer from a square-root transfer bound (Awasthi et al., 2022b). Our Theorem 4.1 shows that by aggregating pairwise margins

using our novel linear-core base, the multi-class Linear-Core surrogate strictly improves this to a linear bound.

- **Structured Prediction:** Previous work on structured sum-exponential surrogates (Mao et al., 2023e) yields a square-root rate. Our Theorem 5.1 demonstrates that the structured Linear-Core surrogate elevates this to a fast linear rate.

To our knowledge, Linear-Core Surrogates are the first explicit family of loss functions to simultaneously guarantee $C^1/C^2$ smoothness and strict linear $\mathcal{H}$-consistency across binary, multi-class, and structured domains.

## 5.3. Optimization and Computational Efficiency

**Optimization Guarantees.** While the standard Hinge loss (and similarly other piecewise linear loss functions) is non-differentiable only at isolated points, this lack of smoothness fundamentally alters the available convergence guarantees. Non-smooth convex optimization relies on sub-gradient methods, which are theoretically limited to a slow convergence rate of $O(1/\sqrt{T})$. In contrast, by establishing that our linear-core surrogates are globally $C^1$ and admit valid second-order approximations (Proposition 3.1), we enable the use of smooth gradient-based optimizers. For smooth convex functions, standard gradient descent guarantees a faster rate of $O(1/T)$, and Nesterov's accelerated gradient methods can achieve the optimal rate of $O(1/T^2)$.

Furthermore, in the structured prediction setting, the loss $\mathsf{L}_{\overline{\Phi}}^{\text{sum}}$ aggregates margins over an exponentially large output space. Here, the non-differentiable *kinks* of a standard Hinge loss form a complex arrangement of hyperplanes rather than a single point, often causing sub-gradient methods to oscillate and stall. To verify this, we conducted a controlled experiment on a synthetic isotropic binary classification problem with orthogonal features.

We minimized both the Hinge loss and the Linear-Core surrogate using Stochastic Gradient Descent (SGD) (Robbins & Monro, 1951) with a fixed learning rate $\eta = 0.1$ and $L_2$ regularization ($\lambda = 0.05$). As empirically demonstrated in Figure 6, the guaranteed $C^1$ smoothness of our surrogates ensures that the gradient magnitude naturally decays near the optimum. This eliminates the chattering phenomenon inherent to the Hinge loss, where non-zero sub-gradients prevent settling, and allows the optimizer to converge linearly to high precision.

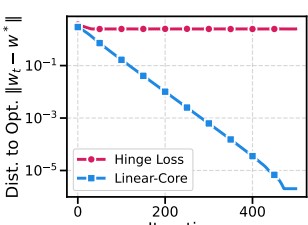

*Figure 6.* Optimization Stability: Linear-Core (Blue) converges linearly; Hinge (Red) stagnates.

**Algorithm 1** Stochastic Gradient Descent for Structured Linear-Core Surrogate

1: **Input:** Training set $S$, step size $\eta$, proposal distributions $\mathcal{D}_1, \mathcal{D}_2$.
2: **Initialize:** $h_0 = 0$.
3: **for** $t = 1$ **to** $T$ **do**
4:   Sample $(x, y)$ uniformly from $S$.
5:   {1. Sample outer label $y'$ based on structural error}
6:   Sample $y' \sim \mathcal{D}_1(\cdot \mid y)$ (e.g., prop. to Hamming distance).
7:   Compute weight $w_1 = \frac{\overline{\ell}(y', y)}{\mathcal{D}_1(y')}$.
8:   {2. Sample inner label $y''$ for comparison}
9:   Sample $y'' \sim \mathcal{D}_2(\cdot \mid y')$ (e.g., uniform neighbor).
10:   Compute weight $w_2 = \frac{1}{\mathcal{D}_2(y'')}$.
11:   {3. Compute Gradient Estimator}
12:   $m \leftarrow h_{t-1}(x, y'') - h_{t-1}(x, y')$
13:   $g_t \leftarrow w_1 w_2 \overline{\Phi}'(m) \cdot (\nabla h(x, y'') - \nabla h(x, y'))$
14:   **Update:** $h_t \leftarrow h_{t-1} - \eta g_t$
15: **end for**
16: **Output:** $h_T$

**Computational Efficiency via Stochastic Sampling.** In the work of Mao et al. (2023e), the computational tractability of the structured sum-exponential loss relies on the algebraic homomorphism of the exponential function, $\exp(u + v) = \exp(u)\exp(v)$. This property allows the sum over exponentially many structures to be decomposed into local factors, enabling exact gradient computation via dynamic programming algorithms (e.g., Forward-Backward or Sum-Product) in polynomial time. Our proposed Structured Linear-Core Surrogates do not satisfy this multiplicative property due to their piecewise definition and linear core. Consequently, exact computation of the full sum loss over $\mathcal{Y} \times \mathcal{Y}$ is generally intractable for large structured spaces.

However, we can ensure computational efficiency by exploiting the additive structure of the loss. We interpret the structured sum loss as an expectation over pairs of labels $(y', y'')$. Specifically, we can rewrite the gradient update as an expectation under a sampling distribution $\mathcal{D}$:
$\nabla \mathsf{L}_{\overline{\Phi}}^{\text{sum}}(h, x, y) = \mathbb{E}_{y' \sim \mathcal{D}_1, y'' \sim \mathcal{D}_2}\big[ \frac{\overline{\ell}(y', y)}{\mathbb{P}_{\mathcal{D}}(y', y'')} \overline{\Phi}'(h(x, y'') - h(x, y'))\nabla(h(x, y'') - h(x, y'))\big]$. By constructing an unbiased estimator of the gradient using Monte Carlo sampling of pairs $(y', y'')$, we reduce the per-iteration computational complexity from $O(|\mathcal{Y}|^2)$ to $O(L)$ (the cost of sampling and embedding a single structure of length $L$). For the outer summation, since $\overline{\ell}(y', y)$ typically decomposes over the structure (e.g., Hamming distance), we can efficiently sample $y'$ from a proposal distribution proportional to the structural error (or simply uniformly with importance weights). For the inner summation, we sample $y''$ via a simple proposal distribution (e.g., uniform or local perturbation).

Crucially, unlike non-smooth structured losses which require solving a global inference problem (Loss-Augmented Inference) at every step, our approach requires only forward sampling. This makes each iteration extremely fast and trivial to parallelize. Furthermore, the smoothness of $\overline{\Phi}$ ensures that the variance of the gradient estimates remains bounded, preserving the convergence guarantees of Stochastic Gradient Descent (SGD).

**Variance of the Stochastic Gradient.** A potential drawback of replacing exact inference with stochastic sampling is the introduction of gradient noise. If the variance of the stochastic gradient were to scale with the size of the output space $|\mathcal{Y}|$, the convergence rate would degrade for large-vocabulary tasks. We show that, remarkably, the variance of our estimator depends only on the number of samples $K$ and the feature radius $R$, and is *independent of* $|\mathcal{Y}|$.

**Theorem 5.3** (Variance Bound for Stochastic Gradients)**.** *Let $\ell(\mathbf{w})$ be the Linear-Core surrogate loss. Let $\widehat{\nabla}\ell(\mathbf{w})$ be the stochastic gradient estimator constructed using a mini-batch of $K$ negative samples $\{\mathbf{y}_k\}_{k=1}^K$ drawn uniformly from $\mathcal{Y}$. Assume the feature map is bounded such that $\|\phi(\mathbf{x}, \mathbf{y})\|_2 \leq R$ for all $\mathbf{x}, \mathbf{y}$. Then, the variance of the estimator is bounded by:* $\mathbb{E}\big[\big\|\widehat{\nabla}\ell(\mathbf{w}) - \nabla\ell(\mathbf{w})\big\|_2^2\big] \leq \frac{4R^2}{K}$.

*Proof Sketch.* The gradient estimator is an average of $K$ independent terms $\mathbf{g}_k$. Since the Linear-Core surrogate is 1-Lipschitz (the derivative is bounded by 1), the norm of any single gradient term is bounded by $\|\mathbf{g}_k\| \leq 1 \cdot \text{diam}(\phi) \leq 2R$. By properties of variance for independent bounded variables, $\text{Var}(\widehat{\nabla}) \leq \frac{1}{K}\sup\|\mathbf{g}\|^2 \leq \frac{4R^2}{K}$. Crucially, this bound relies only on the feature geometry $R$, not the cardinality $|\mathcal{Y}|$. $\square$

**Remark (Contrast with Sampled Softmax).** While techniques like Sampled Softmax allow for $O(1)$ updates, they approximate the Log-Likelihood objective, which suffers from slower square-root consistency rates (see Table 1). Our Stochastic Linear-Core approach is unique in that it combines $O(1)$ sampling efficiency with the fast linear consistency rates of margin-based losses.

### 5.4. Empirical Validation: Sequence Tagging Efficiency

We empirically validate the efficiency of our method on sequence tagging tasks. Figure 7 compares the training time per batch against the Structured SVM (SSVM) as the vocabulary size $|\mathcal{Y}|$ increases. While SSVM scales quadratically ($O(|\mathcal{Y}|^2)$) due to the Viterbi bottleneck, our Linear-Core surrogate with stochastic sampling (Algorithm 1) maintains constant throughput ($O(1)$), achieving a 23× speedup at $|\mathcal{Y}| = 400$. Full experimental details and additional convergence analyses are provided in Appendix E.

### 5.5. Real-World Efficiency: Fine-Grained POS Tagging

To rigorously quantify the computational advantage of our method, we conducted a stress test designed to expose the quadratic bottleneck of the CRF. We used the *Penn Treebank (PTB)* Part-of-Speech tagging dataset (Marcus et al., 1993) but simulated a fine-grained tagging task by artificially inflating the tag set size to $|\mathcal{Y}| = 4000$. This simulates complex morpho-syntactic tagging or open-domain sequence labeling tasks where the label space is large.

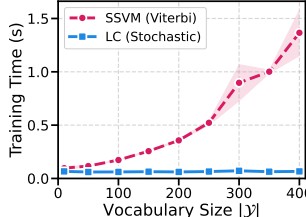

*Figure 7.* Scalability: SSVM $O(|\mathcal{Y}|^2)$ vs. Linear-Core $O(1)$.

**Setup.** To ensure that our efficiency claims hold perfectly regardless of the underlying neural backbone, we compared methods across two different architectures: a standard Bidirectional LSTM (Hochreiter & Schmidhuber, 1997) and a modern BERT-based Transformer backbone (Devlin et al., 2019).

The baselines (BiLSTM-CRF and BERT-CRF) use the respective backbone followed by a Conditional Random Field (CRF) layer (Lafferty et al., 2001). These models minimize the negative log-likelihood of the correct tag sequence, computing gradients exactly via the Forward-Backward algorithm with a time complexity of $O(L|\mathcal{Y}|^2)$ per sequence. We compare these against our proposed BiLSTM-Linear-Core and BERT-Linear-Core, which use the same architectures but minimize the Linear-Core surrogate loss using the stochastic sampling algorithm (Algorithm 1) described in Section 5.3. Crucially, this approach reduces the time complexity to $O(L)$.

For the BiLSTM models, we used an embedding dimension of 128 and a hidden dimension of 256. Training was performed using SGD with momentum 0.9 (Nesterov, 1983) (and AdamW (Loshchilov & Hutter, 2019) for BERT). Critically, we restricted the batch size to $B = 8$. This was necessitated by the CRF baselines, which incur a prohibitive memory cost due to storing the computation graph for $4000 \times 4000$ transition interactions at every sequence step. Our method, having $O(L)$ memory complexity, could theoretically support much larger batches, but we maintained the same batch size for a fair, controlled comparison. To ensure the bottleneck was strictly computational, we filtered the dataset to include only sequences with length $L \geq 100$.

**Results.** We measured the wall-clock time to reach a target test accuracy of 83%. As shown in Figure 8, the results are dramatic.

For the BiLSTM architecture, the BiLSTM-CRF (Red) struggles with the computational load. A single training epoch requires 78.4 seconds due to the expensive matrix op-

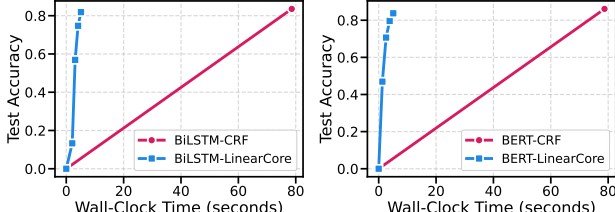

*Figure 8.* Real-world efficiency: Time to accuracy for sequence tagging ($|\mathcal{Y}| = 4000$) using BiLSTM (Left) and BERT (Right) architectures.

erations of the partition function. In contrast, the BiLSTM-Linear-Core (Blue) converges rapidly. By using our stochastic sampling algorithm, it bypasses the quadratic bottleneck entirely, reaching the target accuracy in just 4.5 seconds. Quantitatively, our method achieves a **17.4× speedup** in time-to-accuracy over the exact CRF baseline.

These findings perfectly mirror our results on the modern Transformer backbone, proving that the computational bottleneck lies strictly in the structured loss layer rather than the encoder. The BERT-CRF baseline required 78.7 seconds to reach the target accuracy, remaining bottlenecked by the computation of the partition function over the massive transition space. In contrast, BERT-Linear-Core bypassed the quadratic bottleneck entirely, reaching the target accuracy in just 5.1 seconds, which corresponds to a **15.4× speedup** over the exact CRF inference.

These results confirm that Linear-Core Surrogates enable expressive structured models in large-output domains where traditional CRFs are computationally intractable. While standard English POS tagging has small tag sets, many morphologically rich languages (Finnish, Turkish, Arabic) have tag sets scaling into the thousands. Our simulation with $|\mathcal{Y}| = 4000$ shows that Linear-Core surrogates unlock efficient structured prediction for such complex tasks.

## 6. Conclusion

We introduced Linear-Core Surrogates, the first explicit family of smooth convex loss functions that provably achieve linear $\mathcal{H}$-consistency bounds. By resolving the long-standing dichotomy between smoothness and fast consistency rates, these surrogates simultaneously enable the fast $O(1/T)$ optimization convergence of differentiable losses and the optimal linear transfer rate previously available only to non-smooth losses like the Hinge loss. We established these guarantees progressively across binary, multi-class, and structured prediction, demonstrating the generality of the construction. Beyond their theoretical significance, Linear-Core Surrogates yield practical benefits: natural robustness to instance-dependent label noise in multi-class classification, and a stochastic gradient algorithm for structured prediction that bypasses the quadratic complexity of exact inference, delivering massive speedups over standard baselines.

## Impact Statement

This paper presents work whose goal is to advance the field of Machine Learning. There are many potential societal consequences of our work, none which we feel must be specifically highlighted here.

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

# Contents of Appendix

# A. Related Work

**Comparison with Convolutional Fenchel–Young Losses**     Cao et al. (2025) also address the trade-off between smoothness and linear convergence rates by introducing *Convolutional Fenchel–Young losses*, defined via the infimal convolution of a generalized negentropy regularizer and the target Bayes error. While their framework is theoretically interesting, it differs from ours in several important respects.

*(1) Implicit vs. explicit construction.*  The Fenchel–Young losses are defined *implicitly* as the solution to a variational optimization problem. An explicit closed-form expression is derived only in the binary classification setting, where the resulting loss profile effectively stitches a linear segment to a smooth tail—structurally matching our construction. Indeed, in this special case their explicit binary loss is a particular instance of our more general Linear-Core family (parameterized by an arbitrary smooth base $\Phi$ with $\Phi'(0) > 0$). In the multi-class and structured settings, their framework does not yield explicit loss functions amenable to direct evaluation; each gradient step requires solving an auxiliary optimization problem. In contrast, our surrogates are fully explicit and admit $O(1)$ evaluation and gradient computation without auxiliary solvers.

*(2) Decoding and predictor form.*  The framework of Cao et al. (2025) relies on specific link functions (derived from gradients of the convex conjugate) to decode scores into predictions, departing from the standard argmax rule used in virtually all deep learning applications. Our Linear-Core surrogates retain the standard argmax decoding, $\mathsf{h}(x) = \mathrm{argmax}_{y \in \mathcal{Y}} h(x, y)$, and serve as direct, smooth replacements for the Hinge loss in any existing architecture.

*(3) Structured prediction.*  To the best of our knowledge, the framework of Cao et al. (2025) does not address the computational challenges of structured prediction where the output space $\mathcal{Y}$ is exponentially large. In such settings, generic discrete bounds are insufficient as they may hide dependencies on $|\mathcal{Y}|$ that make computation intractable. Our work explicitly extends linear $\mathcal{H}$-consistency analysis to the structured setting (Theorem 5.1), proves that convexity and smoothness are preserved, and introduces a stochastic gradient algorithm (Algorithm 1) that reduces the per-step complexity from $O(|\mathcal{Y}|^2)$ to $O(1)$ per position.

*(4) Stronger $\mathcal{H}$-consistency guarantees.*  The regret bounds established by Cao et al. (2025) are limited to excess loss bounds with respect to the family of all measurable functions (Bayes consistency). Such bounds guarantee convergence to the Bayes optimal classifier only in the non-parametric limit. In contrast, we provide *linear $\mathcal{H}$-consistency bounds* (Awasthi et al., 2022b; Mao et al., 2023a;b;c;d;f; 2024b;c;d;e;f;g;h; Mohri et al., 2024; Cortes et al., 2024; 2025a;b; Mao et al., 2025a;b;c; Mao, 2025; Zhong, 2025; DeSalvo et al., 2025; Cortes et al., 2026a;b; Montreuil et al., 2025a;b; 2026a;b;c;d;e;f;g; Mohri & Zhong, 2026a;b;c; Mohri et al., 2026). These bounds explicitly relate the estimation error of the surrogate within a restricted hypothesis set $\mathcal{H}$ to the estimation error of the target loss. This is a strictly stronger guarantee that remains valid even for misspecified models (where $\mathcal{H}$ does not contain the Bayes optimal classifier), ensuring that optimizing the smooth Linear-Core Surrogate effectively minimizes the regret against the best possible competitor in $\mathcal{H}$.

**Relationship with the Universal Square-Root Growth Rate**     Our results may appear at first glance to contradict the universal square-root growth rate established by Mao, Mohri, and Zhong (2024a), who proved that $\mathcal{H}$-consistency bounds for "smooth" margin-based surrogates necessarily exhibit a square-root growth rate $\mathcal{T}(t) = \Theta(t^2)$. However, there is no contradiction: the definition of smoothness in their work requires $\Phi$ to be *twice continuously differentiable with $\Phi''(0) > 0$*, i.e., with *non-vanishing curvature at the origin*. This condition is satisfied by all standard smooth losses (logistic, exponential, squared hinge), but it is *violated by design* in our Linear-Core surrogates: our construction guarantees $\overline{\Phi}''(0) = 0$ (or, more precisely, $\overline{\Phi}$ is affine on $[-1, 1]$, so the second derivative vanishes on the entire linear core). This vanishing curvature is precisely the mechanism that allows the transformation $\mathcal{T}(t)$ to grow linearly rather than quadratically near zero (Lemma 3.3), thereby escaping the square-root barrier. In other words, the result of Mao et al. (2024a) characterizes a fundamental limitation of losses with non-zero curvature at the decision boundary, and our Linear-Core surrogates circumvent this limitation by design.

**Contrast with Huber Smoothing**     It is important to distinguish our approach from standard Huber smoothing (Huber, 1964). In the context of classification, Huber-style smoothing typically replaces the non-differentiable "kink" of the Hinge loss (or the region near the decision boundary) with a quadratic segment to ensure differentiability. While effective for optimization, replacing the linear segment with a quadratic destroys the local linearity at the origin. This structural change generally degrades the $\mathcal{H}$-consistency bound from a fast linear rate to a slower square-root rate (similar to the Squared-Hinge loss) (Awasthi et al., 2022a). In contrast, our method explicitly *preserves* the linearity at the origin (specifically in the interval $[-1, 1]$). This retention of the "linear core" is precisely what enables the fast rates derived in Theorem 3.4 and

Corollary 3.6, while the smoothing is applied only to the tails to facilitate gradient-based optimization.

# B. Proof of Proposition 3.1

**Proposition 3.1** (Convexity and Smoothness). *Let $\phi \in \{\overline{\Phi}, \widetilde{\Phi}\}$. The Linear-Core surrogates (Binary $\phi$, Multi-class $\ell_\phi^{\mathrm{sum}}$, and Structured $\mathsf{L}_\phi^{\mathrm{sum}}$ defined in Sections 4 and 5) are globally convex and continuously differentiable ($C^1$). Furthermore, if the base $\Phi$ satisfies $\Phi''(0) = 0$, they are twice continuously differentiable ($C^2$).*

*Proof.* We prove the convexity and smoothness for the binary surrogates first, and then extend to the multi-class and structured cases.

## 1. Binary Symmetric Surrogate $\overline{\Phi}$

*Convexity:* Recall the definition of $\overline{\Phi}$:

$$\overline{\Phi}(u) = \begin{cases} -u + 1 + \dfrac{\Phi(0)}{\Phi'(0)}, & -1 \le u \le 1, \\ \dfrac{\Phi(1-u)}{\Phi'(0)}, & u > 1, \\ \dfrac{\Phi(-1-u)}{\Phi'(0)} + 2, & u < -1. \end{cases}$$

Each branch is convex on its own interval:

- On $(-1, 1)$, $\overline{\Phi}$ is linear, hence convex.

- On $(1, \infty)$, $u \mapsto 1 - u$ is affine. Since $\Phi$ is convex, $u \mapsto \Phi(1-u)$ is convex. Positive scaling by $1/\Phi'(0)$ preserves convexity.

- On $(-\infty, -1)$, similarly, the composition with the affine map $u \mapsto -1 - u$ is convex.

It remains to check the junctions $u = \pm 1$. A continuous piecewise $C^1$ function is convex if the derivative is non-decreasing, which requires $\overline{\Phi}'_-(u_0) \le \overline{\Phi}'_+(u_0)$ at any junction $u_0$. Here, $\overline{\Phi}'(u) = -1$ on $(-1, 1)$. For $u > 1$, $\overline{\Phi}'(u) = -\frac{\Phi'(1-u)}{\Phi'(0)}$. As $u \to 1^+$, $1 - u \to 0^-$, so $\overline{\Phi}'_+(1) = -\frac{\Phi'(0)}{\Phi'(0)} = -1$. This matches $\overline{\Phi}'_-(1) = -1$. For $u < -1$, $\overline{\Phi}'(u) = -\frac{\Phi'(-1-u)}{\Phi'(0)}$. As $u \to -1^-$, $-1 - u \to 0^+$, so $\overline{\Phi}'_-(-1) = -\frac{\Phi'(0)}{\Phi'(0)} = -1$. This matches $\overline{\Phi}'_+(-1) = -1$. Thus, the derivative is continuous everywhere and $\overline{\Phi}$ is convex.

*Smoothness ($C^1$):* As shown above, the one-sided derivatives match at $u = \pm 1$. Since $\Phi'$ is continuous on the outer intervals, $\overline{\Phi}'$ is continuous everywhere. Thus $\overline{\Phi} \in C^1(\mathbb{R})$.

*Smoothness ($C^2$):* For $u > 1$, $\overline{\Phi}''(u) = \frac{\Phi''(1-u)}{\Phi'(0)}$. For $u < -1$, $\overline{\Phi}''(u) = \frac{\Phi''(-1-u)}{\Phi'(0)}$. On $(-1, 1)$, $\overline{\Phi}''(u) = 0$. Continuity at $u = 1$ requires $\lim_{u \to 1^+} \overline{\Phi}''(u) = 0$, which implies $\lim_{z \to 0^-} \Phi''(z) = 0$. Similarly at $u = -1$, we need $\lim_{z \to 0^+} \Phi''(z) = 0$. Thus, $\overline{\Phi} \in C^2(\mathbb{R})$ if and only if $\lim_{z \to 0} \Phi''(z) = 0$. In particular, if $\Phi \in C^2$ and $\Phi''(0) = 0$, this holds.

## 2. Binary One-Sided Surrogate $\widetilde{\Phi}$

*Convexity:* On $(-\infty, 1]$, $\widetilde{\Phi}$ is linear (convex). On $(1, \infty)$, it matches $\overline{\Phi}$ (convex). At $u = 1$, the derivatives match at $-1$. Thus $\widetilde{\Phi}$ is convex.

*Smoothness ($C^1$ and $C^2$):* Matching derivatives at $u = 1$ implies $\widetilde{\Phi} \in C^1(\mathbb{R})$. For $C^2$, we require $\lim_{u \to 1^+} \widetilde{\Phi}''(u) = 0$, which implies $\lim_{z \to 0^-} \Phi''(z) = 0$.

## 3. Multi-class and Structured Extensions

Let $\phi \in \{\overline{\Phi}, \widetilde{\Phi}\}$. The multi-class loss $\ell_\phi^{\mathrm{sum}}(h, x, y) = \sum_{y' \ne y} \phi(h(x, y) - h(x, y'))$ and structured loss $\mathsf{L}_\phi^{\mathrm{sum}}$ are non-negative linear combinations of terms of the form $\phi(L(h))$, where $L$ is a linear functional of the score vector $h(x, \cdot)$. Since $\phi$ is convex and $L$ is linear, $\phi \circ L$ is convex. The sum is therefore convex. Since $\phi$ is $C^1$ (or $C^2$), and $L$ is smooth, the

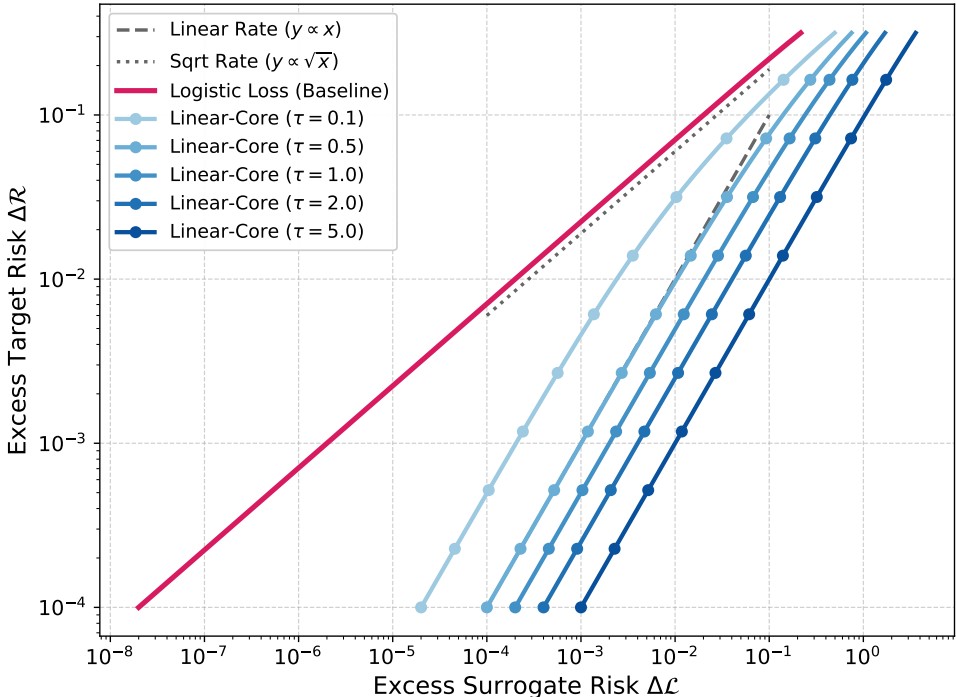

*Figure 9.* Stability of Convergence Rates across Core Thresholds. We compare the excess target error $\Delta\mathcal{R}$ against the excess surrogate error $\Delta\mathcal{L}$ for the generalized Linear-Core surrogate $\overline{\Phi}_\tau$ with varying threshold widths $\tau$. Regardless of whether the linear core is narrow ($\tau = 0.1$) or wide ($\tau = 5.0$), all Linear-Core variants (Blue lines) maintain a strict linear convergence rate parallel to the $y \propto x$ asymptote. In contrast, the Logistic loss (Red) exhibits the slower square-root rate ($y \propto \sqrt{x}$).

composition is $C^1$ (or $C^2$). Thus, the multi-class and structured losses inherit the smoothness properties of the base scalar surrogate. $\qquad\square$

## C. Stability Analysis

To verify that the fast linear rates are not an artifact of the specific interval $[-1, 1]$, we consider a generalized family of Linear-Core surrogates parameterized by a threshold $\tau > 0$. We define the generalized surrogate $\overline{\Phi}_\tau$ by stitching the linear core on $[-\tau, \tau]$ to the smooth tail:

$$\overline{\Phi}_\tau(u) = \begin{cases} -u + \tau + \dfrac{\Phi(0)}{\Phi'(0)}, & -\tau \leq u \leq \tau, \\[2mm] \dfrac{\Phi(\tau - u)}{\Phi'(0)}, & u > \tau, \\[2mm] \dfrac{\Phi(-\tau - u)}{\Phi'(0)} + 2\tau, & u < -\tau. \end{cases} \tag{4}$$

Since $\overline{\Phi}_\tau$ is obtained by affine scaling of the argument $u \mapsto u/\tau$ and the function values, it inherits the convexity and smoothness properties of the base $\Phi$ exactly as established in Proposition 3.1. Furthermore, the linear $\mathcal{H}$-consistency bound (Theorem 3.4) extends naturally to $\overline{\Phi}_\tau$. The transformation $\mathcal{T}$ maintains a linear lower bound $\mathcal{T}(t) \geq \frac{1}{\tau} t$ with $\mathcal{T}(0) = 0$, preserving the fast $O(\Delta\mathcal{L})$ convergence rate.

We first analyzed the convergence rates for robust thresholds $\tau \in \{0.1, 0.5, 1.0, 2.0, 5.0\}$. As shown in Figure 9, the linear convergence rate is robust to the choice of $\tau$ in this regime. All Linear-Core variants maintain a slope of $1$ (implying $\Delta\mathcal{R} = O(\Delta\mathcal{L})$), standing in sharp contrast to the Logistic loss, which degrades to a slope of $1/2$ (implying $\Delta\mathcal{R} = O(\sqrt{\Delta\mathcal{L}})$). This confirms that the fast rate is driven by the non-vanishing curvature at the origin provided by the linear segment.

However, we also investigated the limit case where the linear core vanishes ($\tau \to 0$). We tested microscopic thresholds $\tau \in \{10^{-1}, \dots, 10^{-5}\}$. As shown in Figure 10, as $\tau$ approaches zero, the surrogate $\overline{\Phi}_\tau$ effectively reverts to a standard

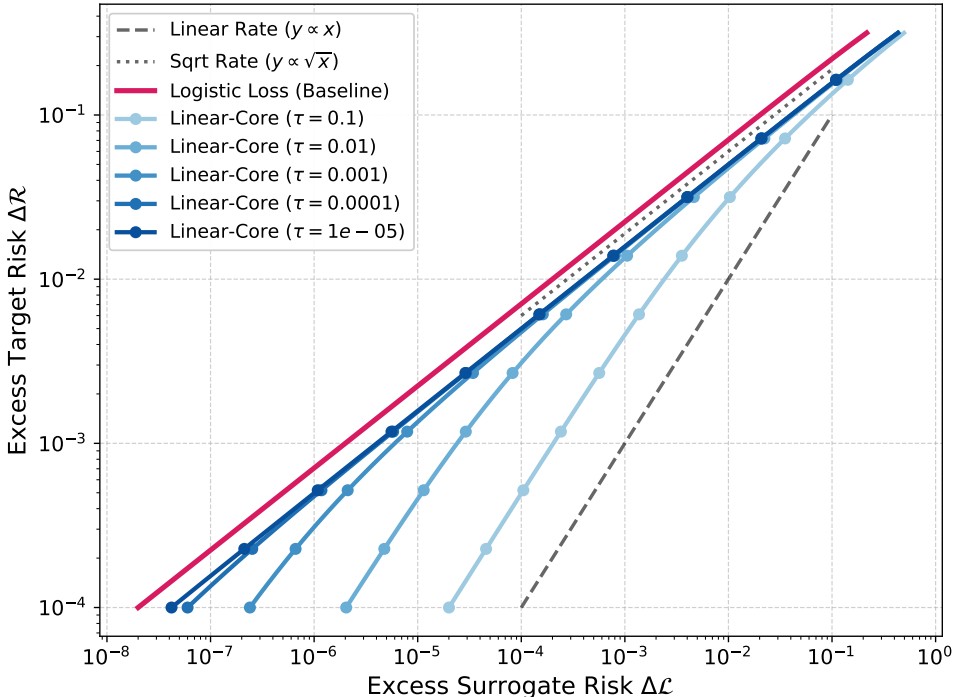

*Figure 10.* Degradation of Convergence Rates as Core Vanishes ($\tau \to 0$). We analyze the convergence behavior for decreasing thresholds $\tau \in \{10^{-1}, \ldots, 10^{-5}\}$. As the linear core shrinks, the Linear-Core variants (darker blue lines) gradually lose their linear acceleration. For the smallest threshold ($\tau = 10^{-5}$), the curve fully aligns with the Logistic Loss (Red), reverting to the standard square-root rate ($y \propto \sqrt{x}$). This confirms that the linear core is the necessary structural element for fast rates.

smooth loss function. Consequently, the acceleration vanishes: the curves for the smallest thresholds (e.g., $\tau = 10^{-5}$) align with the Logistic baseline, exhibiting the slower square-root convergence rate ($y \propto \sqrt{x}$). This demonstrates that the acceleration is strictly dependent on the presence of a non-negligible linear component; when this component is removed, the fast rate is lost.

## D. Multi-Class Examples

To illustrate the behavior of the multi-class linear-core surrogates, we present two concrete instances derived from the logistic and exponential losses. The resulting loss surfaces are visualized in Figure 3 (main text), plotting the loss $\sum_{y' \neq y} \overline{\Phi}(h(x, y) - h(x, y'))$ as a function of the margins $m_1 = h(x, y) - h(x, y_1)$ and $m_2 = h(x, y) - h(x, y_2)$.

**Logistic Linear-Core Surrogate.** Let $\Phi(u) = \log(1 + e^u)$. We have $\Phi(0) = \log 2$ and $\Phi'(0) = 1/2$. The corresponding surrogate $\overline{\Phi}_{\log}$ is given by:

$$\overline{\Phi}_{\log}(u) = \begin{cases} -u + 1 + 2\log 2, & -1 \leq u \leq 1, \\ 2\log(1 + e^{1-u}), & u > 1, \\ 2\log(1 + e^{-1-u}) + 2, & u < -1. \end{cases}$$

In the three-class setting, the total loss is the sum of the surrogates applied to each margin:

$$\ell_{\overline{\Phi}_{\log}}^{\text{sum}}(m_1, m_2) = \overline{\Phi}_{\log}(m_1) + \overline{\Phi}_{\log}(m_2).$$

This function behaves linearly for small margins and transitions smoothly to the scaled logistic tail for large positive margins.

**Exponential Linear-Core Surrogate.** Let $\Phi(u) = e^u$. We have $\Phi(0) = 1$ and $\Phi'(0) = 1$. The surrogate $\overline{\Phi}_{\exp}$ is:

$$\overline{\Phi}_{\exp}(u) = \begin{cases} -u + 2, & -1 \le u \le 1, \\ e^{1-u}, & u > 1, \\ e^{-1-u} + 2, & u < -1. \end{cases}$$

Similarly, the total loss for the three-class case is given by:

$$\ell_{\overline{\Phi}_{\exp}}^{\text{sum}}(m_1, m_2) = \overline{\Phi}_{\exp}(m_1) + \overline{\Phi}_{\exp}(m_2).$$

This creates a loss that is linear in the central region $[-1, 1]$ and decays exponentially for large positive margins, offering a robust alternative to the standard sum-exponential loss.

# E. Empirical Validation: Efficiency in Sequence Tagging – Full Details

To empirically validate the computational efficiency claims discussed in Section 5.3, we compare our method against a standard baseline on a sequence tagging task.

A critical practical limitation of standard structured prediction methods like the Structured SVM (SSVM) is the computational cost of the training loop. The SSVM objective, $\max_{y' \ne y} \max(0, \ell(y', y) - (h(x, y) - h(x, y')))$, requires solving a *loss-augmented inference* problem (finding the "most violated constraint") at every gradient step. For sequence tagging tasks, the standard evaluation metric is the Hamming loss, defined as $\ell(y, y') = \frac{1}{L} \sum_{j=1}^{L} \mathbb{1}_{y'_j \ne y_j}$, where $L$ is the sequence length and $y_j$ denotes the label at the $j$-th position. Optimizing the SSVM with this loss necessitates running the Viterbi algorithm (Viterbi, 2003), which scales as $O(L|\mathcal{Y}|^2)$ per sample and is difficult to parallelize on modern hardware.

In contrast, our Structured Linear-Core Surrogate exploits the additive structure described in Algorithm 1 (Section 5.3), where the loss is evaluated by aggregating local margins rather than solving a global maximization problem. This allows the objective to be optimized using simple stochastic sampling, avoiding the sequential inference bottlenecks inherent to the SSVM.

**Setup.** We consider a synthetic sequence labeling task with sequence length $L = 20$, label set size $|\mathcal{Y}| = 200$, and input dimension $d = 20$. We generate 1,000 training sequences using a linear Hidden Markov Model (Rabiner, 2002) with strong random transition potentials to ensure that structural dependencies are significant. We train a linear neural sequence model following the architecture of Collobert et al. (2011), consisting of a linear projection for unary scores and a learnable transition matrix. We optimize the model using Stochastic Gradient Descent (SGD) (Robbins & Monro, 1951) with a fixed learning rate $\eta = 0.01$ and a batch size of 1. We compare the wall-clock training time required to reach a target test error for the Structured SVM (Tsochantaridis et al., 2005) (implemented with an exact Viterbi solver) against our Structured Linear-Core Surrogate (implemented with the stochastic sampling strategy).

**Convergence Speed.** Figure 11 plots the test Hamming error against wall-clock training time. The Structured SVM (Red) suffers from the high overhead of the Viterbi oracle, resulting in slow convergence in real time, requiring over 200 seconds to minimize the error. Our Linear-Core Surrogate (Blue), using the smooth one-sided logistic tail $\widetilde{\Phi}_{\log}(u)$ and efficient additive decomposition, demonstrates a dramatic speedup. It converges to the optimal error rate almost immediately (within the first few seconds), validating that our method offers both the theoretical benefits of consistency and the practical advantage of computational efficiency in structured domains.

**Scalability Analysis.** We further investigate the impact of vocabulary size on training throughput. As noted previously, solving the loss-augmented inference problem via Viterbi (for SSVM) imposes a quadratic dependency $O(L|\mathcal{Y}|^2)$. As shown in Figure 12 (Red curve), this quadratic complexity makes training prohibitively slow as the vocabulary size grows; increasing $|\mathcal{Y}|$ from 100 to 400 results in a nearly $8\times$ increase in training time for the Structured SVM.

In contrast, the Linear-Core surrogate proposed in this work is differentiable everywhere. This smoothness property fundamentally changes the optimization landscape: instead of solving a combinatorial maximization problem (argmax) at every step, we can estimate the gradient as an expectation over the label space. This allows us to use *unbiased stochastic sampling* to approximate the gradient, decoupling the computational cost from the size of the output space. As shown in Figure 12 (Blue curve), our method maintains a constant throughput regardless of vocabulary size. At $|\mathcal{Y}| = 400$, the

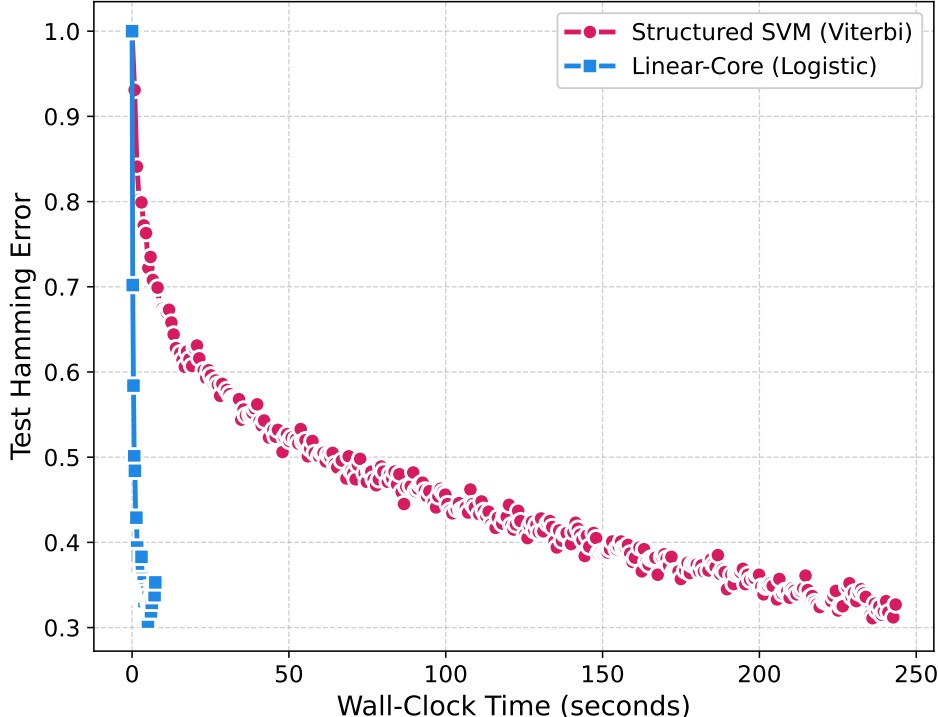

*Figure 11.* Test Error vs. Wall-Clock Time for Sequence Tagging. We compare a Structured SVM (Red Circles) against our Logistic Linear-Core Surrogate (Blue Squares) on a synthetic sequence labeling task ($L = 20, |\mathcal{Y}| = 200$). The Structured SVM is bottlenecked by the sequential Viterbi algorithm required for every gradient update. In contrast, our method processes updates significantly faster due to the efficiency of the stochastic sampling approach, achieving comparable test error in a fraction of the wall-clock time (e.g., reaching $< 0.35$ error in under 10 seconds versus $\approx 250$ seconds for SSVM).

Linear-Core surrogate achieves a **23×** **speedup** over the SSVM. This confirms that our approach enables efficient linear-rate training on large-scale structured problems where traditional max-oracle methods are intractable.

It is important to note that *Conditional Random Fields (CRF)* (Lafferty et al., 2001), the standard probabilistic approach for sequence modeling, shares the same computational bottleneck as SSVM. The gradient of the CRF log-likelihood requires computing marginal probabilities via the Forward-Backward algorithm, which also scales as $O(L|\mathcal{Y}|^2)$.

Furthermore, while widely used approaches such as CRF and SSVM are natural, recent theoretical analysis has shown that their associated loss functions are not Bayes-consistent with respect to discrete target losses, such as the Hamming loss (Mao et al., 2023e). Consequently, these methods inherently cannot be supported by the strong linear $\mathcal{H}$-consistency bounds that we establish for Linear-Core surrogates in Theorem 5.1.

## F. Definitions for Consistency Proofs

For a loss function $\ell$, we define the *conditional error* of a hypothesis $h \in \mathcal{H}$ at a point $x \in \mathcal{X}$ as

$$\mathcal{C}_\ell(h, x) = \sum_{y \in \mathcal{Y}} p(y \mid x)\, \ell(h, x, y),$$

where $p(y \mid x) = \mathcal{D}(Y = y \mid X = x)$ is the conditional probability of $y$ given $x$. The *best-in-class conditional error* is defined as

$$\mathcal{C}_\ell^*(\mathcal{H}, x) = \inf_{h \in \mathcal{H}} \mathcal{C}_\ell(h, x).$$

The *conditional regret* is the difference between the conditional error and the best-in-class conditional error:

$$\Delta\mathcal{C}_{\ell,\mathcal{H}}(h, x) = \mathcal{C}_\ell(h, x) - \mathcal{C}_\ell^*(\mathcal{H}, x).$$

The generalization error can be expressed as the expectation of the conditional error: $\mathcal{E}_\ell(h) = \mathbb{E}_x[\mathcal{C}_\ell(h, x)]$.

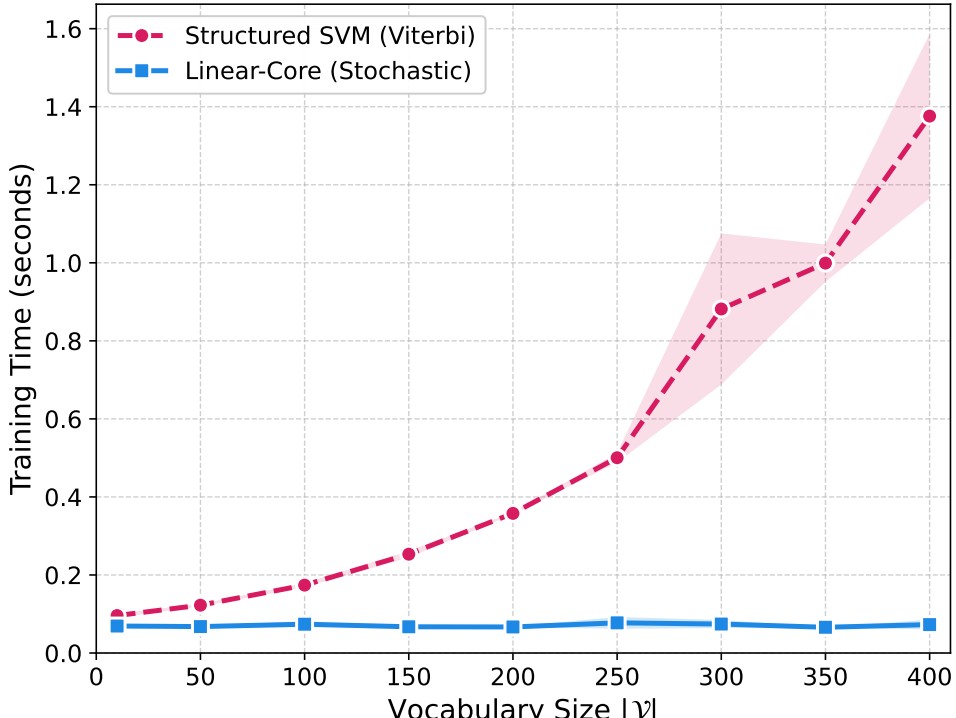

*Figure 12.* Scalability with Vocabulary Size. We compare the wall-clock training time per batch of the Structured SVM (using Viterbi inference) against our Linear-Core Surrogate (using stochastic sampling) as the label vocabulary size $|\mathcal{Y}|$ increases. The SSVM (Red) suffers from the quadratic complexity of dynamic programming ($O(L|\mathcal{Y}|^2)$). In contrast, the Linear-Core surrogate enables unbiased gradient estimation via sampling, resulting in near-constant scaling ($O(L)$). At $|\mathcal{Y}| = 400$, our method achieves a **23× speedup**, reducing the computational bottleneck of large-scale structured learning.

## G. Proofs for Binary Consistency Bounds

### G.1. Proof of Theorem 3.2

**Theorem 3.2.** *Let $\mathcal{H}$ be a complete hypothesis set. The transformation $\mathfrak{T}$ can be expressed as follows:*

$$\forall t \in [0,1], \quad \mathfrak{T}(t) = \overline{\Phi}(0) - \inf_{u \in \mathbb{R}} \left( \frac{1-t}{2} \overline{\Phi}(-u) + \frac{1+t}{2} \overline{\Phi}(u) \right).$$

*Proof.* Since $\overline{\overline{\Phi}}$ is convex and differentiable at zero and satisfies $\overline{\overline{\Phi}}'(0) = -1 < 0$, by Mao et al. (2024a, Theorem 4.1), we complete the proof. □

### G.2. Proof of Lemma 3.3

**Lemma 3.3.** *For all $t \in [0,1]$, $\mathfrak{T}(t) \geq t$ and $\mathfrak{T}(0) = 0$.*

*Proof.* By definition,

$$\mathfrak{T}(t) = \overline{\Phi}(0) - \inf_{u \in \mathbb{R}} \left( \frac{1-t}{2} \overline{\Phi}(-u) + \frac{1+t}{2} \overline{\Phi}(u) \right).$$

**Step 1. Lower bound $\mathfrak{T}(t) \geq t$.** On $[-1,1]$, $\overline{\Phi}$ is linear:

$$\overline{\Phi}(u) = -u + 1 + \frac{\Phi(0)}{\Phi'(0)}, \qquad \overline{\Phi}(-u) = u + 1 + \frac{\Phi(0)}{\Phi'(0)}.$$

Thus for any $u \in [-1, 1]$,

$$\frac{1-t}{2} \overline{\Phi}(-u) + \frac{1+t}{2} \overline{\Phi}(u) = \frac{1-t}{2}\left(u + 1 + \frac{\Phi(0)}{\Phi'(0)}\right) + \frac{1+t}{2}\left(-u + 1 + \frac{\Phi(0)}{\Phi'(0)}\right) = \left(\frac{\Phi(0)}{\Phi'(0)} + 1\right) - tu.$$

For $t \in [0, 1]$, this expression is minimized over $u \in [-1, 1]$ at $u = 1$, giving

$$\inf_{u \in \mathbb{R}} \left(\tfrac{1-t}{2} \overline{\Phi}(-u) + \tfrac{1+t}{2} \overline{\Phi}(u)\right) \le \left(\frac{\Phi(0)}{\Phi'(0)} + 1\right) - t.$$

Since $\overline{\Phi}(0) = \frac{\Phi(0)}{\Phi'(0)} + 1$ (because $0 \in [-1, 1]$), we obtain

$$\mathcal{T}(t) \ge \left(\frac{\Phi(0)}{\Phi'(0)} + 1\right) - \left(\left(\frac{\Phi(0)}{\Phi'(0)} + 1\right) - t\right) = t.$$

**Step 2. Exact value at $t = 0$.** When $t = 0$, we have

$$\mathcal{T}(0) = \overline{\Phi}(0) - \inf_{u \in \mathbb{R}} \tfrac{1}{2}\left(\overline{\Phi}(-u) + \overline{\Phi}(u)\right).$$

If $u \in [-1, 1]$, then

$$\tfrac{1}{2}\left(\overline{\Phi}(-u) + \overline{\Phi}(u)\right) = \tfrac{1}{2}\left(u + 1 + \frac{\Phi(0)}{\Phi'(0)} + \left(-u + 1 + \frac{\Phi(0)}{\Phi'(0)}\right)\right) = \frac{\Phi(0)}{\Phi'(0)} + 1.$$

Thus every $u \in [-1, 1]$ attains the value $\frac{\Phi(0)}{\Phi'(0)} + 1$.

For $u > 1$ or $u < -1$, one checks from the outer branches of $\overline{\Phi}$ and convexity that

$$\tfrac{1}{2}\left(\overline{\Phi}(-u) + \overline{\Phi}(u)\right) \ge \frac{\Phi(0)}{\Phi'(0)} + 1,$$

with equality only at the boundary $u = \pm 1$.

Therefore,

$$\inf_{u \in \mathbb{R}} \tfrac{1}{2}\left(\overline{\Phi}(-u) + \overline{\Phi}(u)\right) = \frac{\Phi(0)}{\Phi'(0)} + 1,$$

and the set of minimizers is precisely $[-1, 1]$. Since $\overline{\Phi}(0) = \frac{\Phi(0)}{\Phi'(0)} + 1$, it follows that

$$\mathcal{T}(0) = 0.$$

Combining the two steps proves the claim. $\qquad\square$

### G.3. Proof of Theorem 3.4

**Theorem 3.4** (Linear $\mathcal{H}$-consistency bound). *Let $\mathcal{H}$ be a complete hypothesis set. Then, for all $h \in \mathcal{H}$,*

$$\mathcal{E}_{\ell_{0-1}}(h) - \mathcal{E}^*_{\ell_{0-1}}(\mathcal{H}) + \mathcal{M}_{\ell_{0-1}}(\mathcal{H}) \le \mathcal{E}_{\ell_{\overline{\Phi}}}(h) - \mathcal{E}^*_{\ell_{\overline{\Phi}}}(\mathcal{H}) + \mathcal{M}_{\ell_{\overline{\Phi}}}(\mathcal{H}).$$

*Proof.* By Mao et al. (2024a, Theorem 4.1), we have

$$\forall h \in \mathcal{H}, \quad \mathcal{T}\left(\mathcal{E}_{\ell_{0-1}}(h) - \mathcal{E}^*_{\ell_{0-1}}(\mathcal{H}) + \mathcal{M}_{\ell_{0-1}}(\mathcal{H})\right) \le \mathcal{E}_{\ell_{\overline{\Phi}}}(h) - \mathcal{E}^*_{\ell_{\overline{\Phi}}}(\mathcal{H}) + \mathcal{M}_{\ell_{\overline{\Phi}}}(\mathcal{H}).$$

By Lemma 3.3, since $\mathcal{T}(t) \ge t$ for all $t \in [0, 1]$, it follows that

$$\forall h \in \mathcal{H}, \quad \mathcal{E}_{\ell_{0-1}}(h) - \mathcal{E}^*_{\ell_{0-1}}(\mathcal{H}) + \mathcal{M}_{\ell_{0-1}}(\mathcal{H}) \le \mathcal{E}_{\ell_{\overline{\Phi}}}(h) - \mathcal{E}^*_{\ell_{\overline{\Phi}}}(\mathcal{H}) + \mathcal{M}_{\ell_{\overline{\Phi}}}(\mathcal{H}).$$

This completes the proof. $\qquad\square$

## G.4. Proof of Lemma 3.5

**Lemma 3.5** (Linear bound for one-sided smoothing). *For $t \in [0,1]$, define*

$$\mathcal{T}_{\mathrm{one}}(t) = \widetilde{\Phi}(0) - \inf_{u \in \mathbb{R}} \Big( \tfrac{1-t}{2} \widetilde{\Phi}(-u) + \tfrac{1+t}{2} \widetilde{\Phi}(u) \Big).$$

*Then $\mathcal{T}_{\mathrm{one}}(t) \geq t$ and $\mathcal{T}_{\mathrm{one}}(0) = 0$.*

*Proof.* For $u \in [-1,1]$, $\widetilde{\Phi}(u) = -u + 1 + \Phi(0)/\Phi'(0)$ and $\widetilde{\Phi}(-u) = u + 1 + \Phi(0)/\Phi'(0)$. Thus

$$\tfrac{1-t}{2} \widetilde{\Phi}(-u) + \tfrac{1+t}{2} \widetilde{\Phi}(u) = \Big( \frac{\Phi(0)}{\Phi'(0)} + 1 \Big) - t\,u.$$

Minimizing over $u \in [-1,1]$ gives

$$\inf_{u} \Big( \tfrac{1-t}{2} \widetilde{\Phi}(-u) + \tfrac{1+t}{2} \widetilde{\Phi}(u) \Big) \leq \frac{\Phi(0)}{\Phi'(0)} + 1 - t,$$

and since $\widetilde{\Phi}(0) = \dfrac{\Phi(0)}{\Phi'(0)} + 1$, we obtain $\mathcal{T}_{\mathrm{one}}(t) \geq t$. At $t = 0$, the same calculation shows $\mathcal{T}_{\mathrm{one}}(0) = 0$. $\qquad\square$

## G.5. Proof of Corollary 3.6

**Corollary 3.6** (Linear $\mathcal{H}$-consistency bound for one-sided smoothing). *Let $\mathcal{H}$ be a complete hypothesis set. Then for $\widetilde{\Phi}$, the following linear $\mathcal{H}$-consistency bound holds:*

$$\forall h \in \mathcal{H}, \quad \mathcal{E}_{\ell_{0-1}}(h) - \mathcal{E}^*_{\ell_{0-1}}(\mathcal{H}) + \mathcal{M}_{\ell_{0-1}}(\mathcal{H}) \leq \mathcal{E}_{\ell_{\widetilde{\Phi}}}(h) - \mathcal{E}^*_{\ell_{\widetilde{\Phi}}}(\mathcal{H}) + \mathcal{M}_{\ell_{\widetilde{\Phi}}}(\mathcal{H}).$$

*Proof.* Since $\widetilde{\Phi}$ is convex and differentiable at zero and satisfies the inequality $\widetilde{\Phi}'(0) = -1 < 0$, by Mao et al. (2024a, Theorem 4.1), for complete hypothesis sets, the transformation $\mathcal{T}$ is equal to $\mathcal{T}_{\mathrm{one}}$:

$$\forall h \in \mathcal{H}, \quad \mathcal{T}_{\mathrm{one}}\Big( \mathcal{E}_{\ell_{0-1}}(h) - \mathcal{E}^*_{\ell_{0-1}}(\mathcal{H}) + \mathcal{M}_{\ell_{0-1}}(\mathcal{H}) \Big) \leq \mathcal{E}_{\ell_{\widetilde{\Phi}}}(h) - \mathcal{E}^*_{\ell_{\widetilde{\Phi}}}(\mathcal{H}) + \mathcal{M}_{\ell_{\widetilde{\Phi}}}(\mathcal{H}).$$

By Lemma 3.5, since $\mathcal{T}_{\mathrm{one}}(t) \geq t$ for all $t \in [0,1]$, it follows that

$$\forall h \in \mathcal{H}, \quad \mathcal{E}_{\ell_{0-1}}(h) - \mathcal{E}^*_{\ell_{0-1}}(\mathcal{H}) + \mathcal{M}_{\ell_{0-1}}(\mathcal{H}) \leq \mathcal{E}_{\ell_{\widetilde{\Phi}}}(h) - \mathcal{E}^*_{\ell_{\widetilde{\Phi}}}(\mathcal{H}) + \mathcal{M}_{\ell_{\widetilde{\Phi}}}(\mathcal{H}).$$

This completes the proof. $\qquad\square$

# H. Proofs for Multi-class Consistency Bounds

## H.1. Auxiliary Lemma H.1 and Lemma H.2

**Lemma H.1.** *Assume $\mathcal{H}$ is symmetric and complete. Then, for any $x \in \mathcal{X}$, the best-in-class conditional error and the conditional regret for $\mathsf{L}_{0-1}$ can be expressed as follows:*

$$\mathcal{C}^*_{\mathsf{L}_{0-1},\mathcal{H}}(x) = 1 - \max_{y \in \mathcal{Y}} p(y \mid x)$$

$$\Delta\mathcal{C}_{\mathsf{L}_{0-1},\mathcal{H}}(h,x) = \max_{y \in \mathcal{Y}} p(y \mid x) - p(\mathsf{h}(x) \mid x).$$

*Proof.* By Awasthi et al. (2022b, Lemma 3) and the fact that $\mathsf{H}(x) = \mathcal{Y}$ when $\mathcal{H}$ is symmetric, the proof is complete. $\qquad\square$

**Lemma H.2** (Restricted optimizer for $\overline{\Phi}$ and $\widetilde{\Phi}$ on $[-1,1]$). *For $a,b \geq 0$,*

$$\inf_{u \in [-1,1]} \Big( a\,\overline{\Phi}(-u) + b\,\overline{\Phi}(u) \Big) = \inf_{u \in [-1,1]} \Big( a\,\widetilde{\Phi}(-u) + b\,\widetilde{\Phi}(u) \Big) = (a+b)\frac{\Phi(0)}{\Phi'(0)} + 2\min\{a,b\},$$

*with the infimum attained at $u^* = -1$ if $a \geq b$ and at $u^* = 1$ if $a \leq b$.*

*Proof.* For $u \in [-1, 1]$, the middle branch of $\overline{\Phi}$ and $\widetilde{\Phi}$ gives

$$\overline{\Phi}(-u) = \widetilde{\Phi}(-u) = 1 + u + \frac{\Phi(0)}{\Phi'(0)}, \qquad \overline{\Phi}(u) = \widetilde{\Phi}(-u) = 1 - u + \frac{\Phi(0)}{\Phi'(0)}.$$

Therefore, for $a, b \geq 0$,

$$a\,\overline{\Phi}(-u) + b\,\overline{\Phi}(u) = a\,\widetilde{\Phi}(-u) + b\,\widetilde{\Phi}(u) = (a + b)\left(1 + \frac{\Phi(0)}{\Phi'(0)}\right) + (a - b)u.$$

The right-hand side is affine in $u$, hence minimized on the interval $[-1, 1]$ at an endpoint: at $u^* = -1$ if $a \geq b$, and at $u^* = 1$ if $a \leq b$. Evaluating at these points yields

$$\inf_{u \in [-1,1]} \left(a\,\overline{\Phi}(-u) + b\,\overline{\Phi}(u)\right) = \inf_{u \in [-1,1]} \left(a\,\widetilde{\Phi}(-u) + b\,\widetilde{\Phi}(u)\right)$$

$$= (a + b)\left(1 + \frac{\Phi(0)}{\Phi'(0)}\right) - |a - b|$$

$$= (a + b)\frac{\Phi(0)}{\Phi'(0)} + 2\min\{a, b\}.$$

This proves the claim. $\qquad\qquad\square$

### H.2. Proof of Theorem 4.1

**Theorem 4.1** (Linear $\mathcal{H}$-consistency bound for multi-class linear-core surrogates). *Assume $\mathcal{H}$ is symmetric and complete. Then, for any distribution and any $h \in \mathcal{H}$,*

$$\mathcal{R}_{\mathsf{L}_{0-1}}(h) - \mathcal{R}^*_{\mathsf{L}_{0-1}}(\mathcal{H}) + \mathcal{M}_{\mathsf{L}_{0-1}}(\mathcal{H}) \leq \mathcal{R}_{\ell^{\mathrm{sum}}_{\overline{\Phi}}}(h) - \mathcal{R}^*_{\ell^{\mathrm{sum}}_{\overline{\Phi}}}(\mathcal{H}) + \mathcal{M}_{\ell^{\mathrm{sum}}_{\overline{\Phi}}}(\mathcal{H}),$$

$$\mathcal{R}_{\mathsf{L}_{0-1}}(h) - \mathcal{R}^*_{\mathsf{L}_{0-1}}(\mathcal{H}) + \mathcal{M}_{\mathsf{L}_{0-1}}(\mathcal{H}) \leq \mathcal{R}_{\ell^{\mathrm{sum}}_{\widetilde{\Phi}}}(h) - \mathcal{R}^*_{\ell^{\mathrm{sum}}_{\widetilde{\Phi}}}(\mathcal{H}) + \mathcal{M}_{\ell^{\mathrm{sum}}_{\widetilde{\Phi}}}(\mathcal{H}).$$

*Proof.* Fix $x \in \mathcal{X}$. Let $\phi \in \{\overline{\Phi}, \widetilde{\Phi}\}$. For brevity, let $W_y := p(y \mid x)$. Let $y_{\max} \in \mathrm{argmax}_{y \in \mathcal{Y}} W_y$ and let $\mathsf{h}(x) = \mathrm{argmax}_{y \in \mathcal{Y}} h(x, y)$. If $\mathsf{h}(x) = y_{\max}$, by Lemma H.1, the inequality $\Delta\mathcal{C}_{\mathsf{L}_{0-1}, \mathcal{H}}(h, x) \leq \Delta\mathcal{C}_{\ell^{\mathrm{sum}}_\phi, \mathcal{H}}(h, x)$ holds trivially since the left-hand side is zero. Assume $\mathsf{h}(x) \neq y_{\max}$. The conditional error of the sum loss can be decomposed into a sum of pairwise errors. We have:

$$\mathcal{C}_{\ell^{\mathrm{sum}}_\phi}(h, x) = \sum_{y \in \mathcal{Y}} W_y \sum_{y' \neq y} \phi(h(x, y) - h(x, y'))$$

$$= \frac{1}{2} \sum_{\{y, y'\} \subseteq \mathcal{Y}, y \neq y'} \underbrace{\left[W_y \phi(h(x, y) - h(x, y')) + W(y') \phi(h(x, y') - h(x, y))\right]}_{=: \mathcal{C}_{\{y, y'\}}(h, x)}.$$

We first determine the best-in-class conditional error $\mathcal{C}^*_{\ell^{\mathrm{sum}}_\phi}(\mathcal{H}, x)$. Consider any pair $\{y, y'\}$ with $W_y \geq W(y')$. Minimizing the pairwise term $\mathcal{C}_{\{y, y'\}}(h, x)$ requires the margin $h(x, y) - h(x, y')$ to be optimized (typically driven to a positive value). Generally, pairwise constraints might conflict (e.g., violating the triangle inequality). However, here the "preference" direction for every pair is determined by the order of the scalar probabilities $p(\cdot \mid x)$. Since these probabilities induce a total ordering on $\mathcal{Y}$, the pairwise requirements are transitive and acyclic. Therefore, there is no conflict: one can construct a score vector $h(x, \cdot)$ that satisfies the optimal margin requirements for *all* pairs simultaneously (for instance, by setting scores proportional to the rank of $W_y$). Since $\mathcal{H}$ is complete, such a vector exists in $\mathcal{H}$. Thus, the infimum of the sum is the sum of the infimums:

$$\mathcal{C}^*_{\ell^{\mathrm{sum}}_\phi}(\mathcal{H}, x) = \frac{1}{2} \sum_{\{y, y'\} \subseteq \mathcal{Y}, y \neq y'} \inf_{h \in \mathcal{H}} \mathcal{C}_{\{y, y'\}}(h, x).$$

The conditional regret then decomposes additively:

$$\Delta\mathcal{C}_{\ell^{\mathrm{sum}}_\phi, \mathcal{H}}(h, x) = \frac{1}{2} \sum_{\{y, y'\} \subseteq \mathcal{Y}, y \neq y'} \underbrace{\mathcal{C}_{\{y, y'\}}(h, x) - \inf_{h \in \mathcal{H}} \mathcal{C}_{\{y, y'\}}(h, x)}_{=: \Delta\mathcal{C}_{\{y, y'\}}(h, x)}.$$

Since each pairwise regret term $\Delta\mathcal{C}_{\{y,y'\}}(h,x)$ is non-negative, we can lower bound the total regret by the two terms corresponding to the pair $\{y_{\max}, h(x)\}$. Note that $\Delta\mathcal{C}_{\{y,y'\}}(h,x) = \Delta\mathcal{C}_{\{y',y\}}(h,x)$. Let $y_1 = y_{\max}$ and $y_2 = h(x)$. By definition, $p_{y_1} \geq p_{y_2}$. Also, let $m = h(x, y_1) - h(x, y_2)$. Since $y_2$ is the predicted class, $h(x, y_2) \geq h(x, y_1)$, implying $m \leq 0$. The pairwise regret for $\{y_1, y_2\}$ is:

$$\Delta\mathcal{C}_{\{y_1, y_2\}}(h,x) = p_{y_1}\phi(m) + p_{y_2}\phi(-m) - \inf_{u \in \mathbb{R}}(p_{y_1}\phi(u) + p_{y_2}\phi(-u)).$$

We apply Lemma H.2. Since $[-1,1] \subset \mathbb{R}$, the infimum over $\mathbb{R}$ is upper bounded by the restricted infimum over $[-1,1]$ (attained at the boundary $u = 1$ since $p_{y_1} \geq p_{y_2}$), so:

$$\inf_{u \in \mathbb{R}}(p_{y_1}\phi(u) + p_{y_2}\phi(-u)) \leq (p_{y_1} + p_{y_2})\frac{\Phi(0)}{\Phi'(0)} + 2\min\{p_{y_1}, p_{y_2}\} = (p_{y_1} + p_{y_2})\frac{\Phi(0)}{\Phi'(0)} + 2p_{y_2}.$$

For the first term, we use the property that $\phi$ has slope $-1$ at the origin. By convexity, $\phi(t) \geq \phi(0) - t = (1 + \frac{\Phi(0)}{\Phi'(0)}) - t$. Thus:

$$p_{y_1}\phi(m) + p_{y_2}\phi(-m) \geq p_{y_1}\left[1 + \frac{\Phi(0)}{\Phi'(0)} - m\right] + p_{y_2}\left[1 + \frac{\Phi(0)}{\Phi'(0)} + m\right]$$

$$= (p_{y_1} + p_{y_2})\left(1 + \frac{\Phi(0)}{\Phi'(0)}\right) + (p_{y_2} - p_{y_1})m.$$

Subtracting the minimal error:

$$\Delta\mathcal{C}_{\{y_1, y_2\}}(h,x) \geq (p_{y_1} + p_{y_2})\left(1 + \frac{\Phi(0)}{\Phi'(0)}\right) + (p_{y_2} - p_{y_1})m - \left[(p_{y_1} + p_{y_2})\frac{\Phi(0)}{\Phi'(0)} + 2p_{y_2}\right]$$

$$= (p_{y_1} + p_{y_2}) - 2p_{y_2} + (p_{y_2} - p_{y_1})m$$

$$= (p_{y_1} - p_{y_2}) + (p_{y_2} - p_{y_1})m$$

$$= (p_{y_1} - p_{y_2})(1 - m).$$

By the symmetry, we have $\Delta\mathcal{C}_{\{y_2, y_1\}}(h,x) = \Delta\mathcal{C}_{\{y_1, y_2\}}(h,x) \geq (p_{y_1} - p_{y_2})(1 - m)$. Since $p_{y_1} \geq p_{y_2}$ and $m \leq 0$, we have $(p_{y_1} - p_{y_2}) \geq 0$ and $(1 - m) \geq 1$. Therefore, since each pairwise regret is non-negative, we have:

$$\Delta\mathcal{C}_{\ell_\phi^{\mathrm{sum}}, \mathcal{H}}(h,x) \geq \frac{1}{2}\left(\Delta\mathcal{C}_{\{y_1, y_2\}}(h,x) + \Delta\mathcal{C}_{\{y_2, y_1\}}(h,x)\right) \geq p_{y_1} - p_{y_2} = p(y_{\max} \mid x) - p(h(x) \mid x).$$

By Lemma H.1, this lower bound equals $\Delta\mathcal{C}_{\mathsf{L}_{0-1}, \mathcal{H}}(h,x)$. Finally, taking expectations over $x$ yields the statement of the theorem:

$$\mathcal{R}_{\mathsf{L}_{0-1}}(h) - \mathcal{R}^*_{\mathsf{L}_{0-1}}(\mathcal{H}) + \mathcal{M}_{\mathsf{L}_{0-1}}(\mathcal{H}) \leq \mathcal{R}_{\ell_\phi^{\mathrm{sum}}}(h) - \mathcal{R}^*_{\ell_\phi^{\mathrm{sum}}}(\mathcal{H}) + \mathcal{M}_{\ell_\phi^{\mathrm{sum}}}(\mathcal{H}).$$

$\square$

# I. Proofs for Structured Consistency Bounds

## I.1. Auxiliary Lemma I.1

**Lemma I.1.** *Assume $\mathcal{H}$ is symmetric and complete. Then, for any $x \in \mathcal{X}$, the best-in-class conditional error and the conditional regret for $\mathsf{L}$ can be expressed as follows:*

$$\mathcal{C}^*_{\mathsf{L}, \mathcal{H}}(x) = \min_{y' \in \mathcal{Y}} \sum_{y \in \mathcal{Y}} p(y \mid x)\ell(y', y)$$

$$\Delta\mathcal{C}_{\mathsf{L}, \mathcal{H}}(h,x) = \sum_{y \in \mathcal{Y}} p(y \mid x)\ell(h(x), y) - \min_{y' \in \mathcal{Y}} \sum_{y \in \mathcal{Y}} p(y \mid x)\ell(y', y).$$

*Proof.* By Mao et al. (2023e, Lemma 3) and the fact that $\mathsf{H}(x) = \mathcal{Y}$ when $\mathcal{H}$ is symmetric, the proof is complete. $\square$

## I.2. Proof of Theorem 5.1

**Theorem 5.1** (Linear $\mathcal{H}$-consistency bound for structured prediction). *Assume $\mathcal{H}$ is symmetric and complete. Let $\mathsf{L}$ be the target structured loss defined by $\ell$. Then, for any distribution and any $h \in \mathcal{H}$, the following bounds hold:*

$$\mathcal{R}_{\mathsf{L}}(h) - \mathcal{R}_{\mathsf{L}}^*(\mathcal{H}) + \mathcal{M}_{\mathsf{L}}(\mathcal{H}) \le \mathcal{R}_{\mathsf{L}_{\overline{\Phi}}^{\mathrm{sum}}}(h) - \mathcal{R}_{\mathsf{L}_{\overline{\Phi}}^{\mathrm{sum}}}^*(\mathcal{H}) + \mathcal{M}_{\mathsf{L}_{\overline{\Phi}}^{\mathrm{sum}}}(\mathcal{H}),$$

$$\mathcal{R}_{\mathsf{L}}(h) - \mathcal{R}_{\mathsf{L}}^*(\mathcal{H}) + \mathcal{M}_{\mathsf{L}}(\mathcal{H}) \le \mathcal{R}_{\mathsf{L}_{\widetilde{\Phi}}^{\mathrm{sum}}}(h) - \mathcal{R}_{\mathsf{L}_{\widetilde{\Phi}}^{\mathrm{sum}}}^*(\mathcal{H}) + \mathcal{M}_{\mathsf{L}_{\widetilde{\Phi}}^{\mathrm{sum}}}(\mathcal{H}).$$

*Proof.* The proof for $\overline{\Phi}$ and $\widetilde{\Phi}$ is identical due to the coincidence of the functions on $[-1, 1]$. Let $\phi \in \{\overline{\Phi}, \widetilde{\Phi}\}$. We first establish a pointwise lower bound on the surrogate regret. Fix $x$. Let $p_y = p(y|x)$. The target conditional error is $\mathcal{C}_{\mathsf{L}}(h, x) = \sum_{y \in \mathcal{Y}} p_y \ell(\mathsf{h}(x), y)$. Since $\mathcal{H}$ is complete, by Lemma I.1, the best-in-class target conditional error is $\mathcal{C}_{\mathsf{L}}^*(\mathcal{H}, x) = \inf_{y' \in \mathcal{Y}} \sum_{y \in \mathcal{Y}} p_y \ell(y', y)$. Thus, the target conditional regret is:

$$\Delta\mathcal{C}_{\mathsf{L},\mathcal{H}}(h, x) = \sum_{y \in \mathcal{Y}} p_y \ell(\mathsf{h}(x), y) - \inf_{y' \in \mathcal{Y}} \sum_{y \in \mathcal{Y}} p_y \ell(y', y).$$

Now consider the surrogate loss. The conditional surrogate error is:

$$\begin{aligned}
\mathcal{C}_{\mathsf{L}_\phi^{\mathrm{sum}}}(h, x) &= \sum_{y \in \mathcal{Y}} p_y \sum_{y' \in \mathcal{Y}} \overline{\ell}(y', y) \sum_{y'' \ne y'} \phi(h(x, y') - h(x, y'')) \\
&= \sum_{y' \in \mathcal{Y}} \sum_{y'' \ne y'} \phi(h(x, y') - h(x, y'')) \underbrace{\sum_{y \in \mathcal{Y}} p_y \overline{\ell}(y', y)}_{=:W(y')} \\
&= \frac{1}{2} \sum_{\{y', y''\} \subseteq \mathcal{Y}, y' \ne y''} \underbrace{[W(y')\phi(h(x, y') - h(x, y'')) + W(y'')\phi(h(x, y'') - h(x, y'))]}_{=:\mathcal{C}_{\{y', y''\}}(h, x)}.
\end{aligned}$$

Note that the inner summation $W(y')$ does not depend on $y''$. We first determine the best-in-class conditional error $\mathcal{C}_{\mathsf{L}_\phi^{\mathrm{sum}}}^*(\mathcal{H}, x)$. Consider any pair $\{y', y''\}$ with $W(y') \ge W(y'')$. Minimizing the pairwise term $\mathcal{C}_{\{y', y''\}}(h, x)$ requires the margin $h(x, y') - h(x, y'')$ to be optimized (typically driven to a positive value). Generally, pairwise constraints might conflict (e.g., violating the triangle inequality). However, here the "preference" direction for every pair is determined by the order of the scalar weights $W(\cdot)$. Since these weights induce a total ordering on $\mathcal{Y}$, the pairwise requirements are transitive and acyclic. Therefore, there is no conflict: one can construct a score vector $h(x, \cdot)$ that satisfies the optimal margin requirements for *all* pairs simultaneously (for instance, by setting scores proportional to the rank of $W(y)$). Since $\mathcal{H}$ is complete, such a vector exists in $\mathcal{H}$. Thus, the infimum of the sum is the sum of the infimums:

$$\mathcal{C}_{\mathsf{L}_\phi^{\mathrm{sum}}}^*(\mathcal{H}, x) = \frac{1}{2} \sum_{\{y', y''\} \subseteq \mathcal{Y}, y' \ne y''} \inf_{h \in \mathcal{H}} \mathcal{C}_{\{y', y''\}}(h, x).$$

The conditional regret then decomposes additively:

$$\Delta\mathcal{C}_{\mathsf{L}_\phi^{\mathrm{sum}}, \mathcal{H}}(h, x) = \frac{1}{2} \sum_{\{y', y''\} \subseteq \mathcal{Y}, y' \ne y''} \underbrace{\mathcal{C}_{\{y', y''\}}(h, x) - \inf_{h \in \mathcal{H}} \mathcal{C}_{\{y', y''\}}(h, x)}_{=:\Delta\mathcal{C}_{\{y', y''\}}(h, x)}.$$

Let $y_{\max} \in \operatorname{argmax}_{y \in \mathcal{Y}} W(y)$ and let $\mathsf{h}(x) = \operatorname{argmax}_{y \in \mathcal{Y}} h(x, y)$. If $\mathsf{h}(x) = y_{\max}$, by Lemma I.1, the inequality $\Delta\mathcal{C}_{\mathsf{L},\mathcal{H}}(h, x) \le \Delta\mathcal{C}_{\mathsf{L}_\phi^{\mathrm{sum}}, \mathcal{H}}(h, x)$ holds trivially since the left-hand side is zero. Assume $\mathsf{h}(x) \ne y_{\max}$. Since each pairwise regret term $\Delta\mathcal{C}_{\{y', y''\}}(h, x)$ is non-negative, we can lower bound the total regret by the two terms corresponding to the pair $\{y_{\max}, \mathsf{h}(x)\}$. Note that $\Delta\mathcal{C}_{\{y', y''\}}(h, x) = \Delta\mathcal{C}_{\{y'', y'\}}(h, x)$. Let $y_1 = y_{\max}$ and $y_2 = \mathsf{h}(x)$. By definition, $W(y_1) \ge W(y_2)$. Also, let $m = h(x, y_1) - h(x, y_2)$. Since $y_2$ is the predicted class, $h(x, y_2) \ge h(x, y_1)$, implying $m \le 0$. The pairwise regret for $\{y_1, y_2\}$ is:

$$\Delta\mathcal{C}_{\{y_1, y_2\}}(h, x) = W(y_1)\phi(m) + W(y_2)\phi(-m) - \inf_{u \in \mathbb{R}}(W(y_1)\phi(u) + W(y_2)\phi(-u)).$$

We apply Lemma H.2. Since $[-1, 1] \subset \mathbb{R}$, the infimum over $\mathbb{R}$ is upper bounded by the restricted infimum over $[-1, 1]$ (attained at the boundary $u = 1$ since $W(y_1) \geq W(y_2)$), so:

$$\inf_{u \in \mathbb{R}}(W(y_1)\phi(u) + W(y_2)\phi(-u)) \leq (W(y_1) + W(y_2))\frac{\Phi(0)}{\Phi'(0)} + 2\min\{W(y_1), W(y_2)\}$$

$$= (W(y_1) + W(y_2))\frac{\Phi(0)}{\Phi'(0)} + 2W(y_2).$$

For the first term, we use the property that $\phi$ has slope $-1$ at the origin. By convexity, $\phi(t) \geq \phi(0) - t = (1 + \frac{\Phi(0)}{\Phi'(0)}) - t$. Thus:

$$W(y_1)\phi(m) + W(y_2)\phi(-m) \geq W(y_1)\left[1 + \frac{\Phi(0)}{\Phi'(0)} - m\right] + W(y_2)\left[1 + \frac{\Phi(0)}{\Phi'(0)} + m\right]$$

$$= (W(y_1) + W(y_2))\left(1 + \frac{\Phi(0)}{\Phi'(0)}\right) + (W(y_2) - W(y_1))m.$$

Subtracting the minimal error:

$$\Delta\mathcal{C}_{\{y_1, y_2\}}(h, x) \geq (W(y_1) + W(y_2))\left(1 + \frac{\Phi(0)}{\Phi'(0)}\right) + (W(y_2) - W(y_1))m$$

$$- \left[(W(y_1) + W(y_2))\frac{\Phi(0)}{\Phi'(0)} + 2W(y_2)\right]$$

$$= (W(y_1) + W(y_2)) - 2W(y_2) + (W(y_2) - W(y_1))m$$

$$= (W(y_1) - W(y_2)) + (W(y_2) - W(y_1))m$$

$$= (W(y_1) - W(y_2))(1 - m).$$

By the symmetry, we have $\Delta\mathcal{C}_{\{y_2, y_1\}}(h, x) = \Delta\mathcal{C}_{\{y_1, y_2\}}(h, x) \geq (W(y_1) - W(y_2))(1 - m)$. Since $W(y_1) \geq W(y_2)$ and $m \leq 0$, we have $(W(y_1) - W(y_2)) \geq 0$ and $(1 - m) \geq 1$. Therefore, since each pairwise regret is non-negative, we have:

$$\Delta\mathcal{C}_{\mathsf{L}_\phi^{\mathrm{sum}}, \mathcal{H}}(h, x) \geq \frac{1}{2}\left(\Delta\mathcal{C}_{\{y_1, y_2\}}(h, x) + \Delta\mathcal{C}_{\{y_2, y_1\}}(h, x)\right) \geq W(y_1) - W(y_2).$$

By Lemma I.1, this lower bound equals $\Delta\mathcal{C}_{\mathsf{L}, \mathcal{H}}(h, x)$. Finally, taking expectations over $x$ yields the statement of the theorem:

$$\mathcal{R}_\mathsf{L}(h) - \mathcal{R}_\mathsf{L}^*(\mathcal{H}) + \mathcal{M}_\mathsf{L}(\mathcal{H}) \leq \mathcal{R}_{\mathsf{L}_\phi^{\mathrm{sum}}}(h) - \mathcal{R}_{\mathsf{L}_\phi^{\mathrm{sum}}}^*(\mathcal{H}) + \mathcal{M}_{\mathsf{L}_\phi^{\mathrm{sum}}}(\mathcal{H}).$$

$\square$

## J. Proof of Theorem 5.3

**Theorem 5.3** (Variance Bound for Stochastic Gradients). *Let $\ell(\mathbf{w})$ be the Linear-Core surrogate loss. Let $\widehat{\nabla}\ell(\mathbf{w})$ be the stochastic gradient estimator constructed using a mini-batch of $K$ negative samples $\{\mathbf{y}_k\}_{k=1}^K$ drawn uniformly from $\mathcal{Y}$. Assume the feature map is bounded such that $\|\phi(\mathbf{x}, \mathbf{y})\|_2 \leq R$ for all $\mathbf{x}, \mathbf{y}$. Then, the variance of the estimator is bounded by:*
$\mathbb{E}\left[\left\|\widehat{\nabla}\ell(\mathbf{w}) - \nabla\ell(\mathbf{w})\right\|_2^2\right] \leq \frac{4R^2}{K}.$

*Proof.* Recall that the gradient of the Linear-Core loss for a single example $(\mathbf{x}, \mathbf{y}^*)$ can be written as an expectation:

$$\nabla\ell(\mathbf{w}) = \mathop{\mathbb{E}}_{\mathbf{y} \sim \mathbb{P}(\mathbf{y}|\mathbf{x})}[\Psi'(\cdot)(\phi(\mathbf{x}, \mathbf{y}) - \phi(\mathbf{x}, \mathbf{y}^*))], \tag{5}$$

where $\Psi'$ is the scalar derivative of the surrogate and $\mathbb{P}(\mathbf{y}|\mathbf{x})$ is the sampling distribution (e.g., uniform). Let $\mathbf{g}_k$ be the gradient estimate from a single sample $\mathbf{y}_k$:

$$\mathbf{g}_k = \Psi'(\cdot)(\phi(\mathbf{x}, \mathbf{y}_k) - \phi(\mathbf{x}, \mathbf{y}^*)). \tag{6}$$

Since $\Psi'$ is bounded by 1 (Lipschitz property of the Linear-Core) and the feature norm is bounded by $R$, the norm of any single estimate is bounded:

$$\|\mathbf{g}_k\|_2 = |\Psi'(\cdot)| \cdot \|\phi(\mathbf{x}, \mathbf{y}_k) - \phi(\mathbf{x}, \mathbf{y}^*)\|_2 \leq 1 \cdot (R + R) = 2R. \tag{7}$$

The total estimator is the average $\widehat{\nabla}\ell(\mathbf{w}) = \frac{1}{K} \sum_{k=1}^{K} \mathbf{g}_k$. Using the standard variance property for independent random variables:

$$\mathbb{E}\big[\|\widehat{\nabla}\ell(\mathbf{w}) - \nabla\ell(\mathbf{w})\|_2^2\big] = \frac{1}{K^2} \sum_{k=1}^{K} \mathbb{E}\big[\|\mathbf{g}_k - \nabla\ell(\mathbf{w})\|_2^2\big] \tag{8}$$

$$\leq \frac{1}{K} \sup_{\mathbf{y}} \|\mathbf{g}(\mathbf{y})\|_2^2 \tag{9}$$

$$\leq \frac{(2R)^2}{K} = \frac{4R^2}{K}. \tag{10}$$

Thus, the variance decreases linearly with $K$ and is independent of the cardinality $|\mathcal{Y}|$. $\qquad\square$

## K. Generalization to Non-Decomposable Metrics via Rational Kernels

Our structured prediction analysis in Section 5 centered on the Hamming loss due to its immediate token-level additive decomposability. However, the theoretical and computational advantages of the Linear-Core surrogate extend elegantly to non-decomposable metrics such as exact F1-score and BLEU when viewed through the lens of rational kernels (Cortes et al., 2004).

While BLEU is often considered strictly non-decomposable at the token level, its core mechanism, $n$-gram overlap counts, can be expressed exactly as an instance of a rational kernel computed via weighted finite-state transducers (WFSTs). By expanding the state space of our sequence model to encode length-$(n-1)$ histories, the $n$-gram matching scores $\bar{\ell}(y', y)$ become strictly additively factorizable over the transitions of this expanded automaton.

Consequently, we do not need to rely on high-variance, sequence-level sampling strategies (like REINFORCE). Instead, our unbiased stochastic gradient estimator (Algorithm 1) generalizes directly: we simply construct our proposal distributions $\mathcal{D}_1$ and $\mathcal{D}_2$ over the paths of this expanded WFST. Because the Linear-Core surrogate's consistency and variance bounds (Theorem 5.1 and Theorem 5.3) rely only on the additive aggregation of margins, and not on the specific topology of the state space, both the optimal $O(\Delta\mathcal{L})$ linear transfer rate and the $O(1)$ per-step computational efficiency are rigorously preserved.

