# OpenReview forum: "Linear-Core Surrogates: Smooth Loss Functions with Linear Rates for Classification and Structured Prediction"
_ICML.cc/2026/Conference — ICML 2026 regular_

### Official Review · Reviewer_R1kw · 2026-02-27

**Soundness:** 2
**Presentation:** 2
**Significance:** 2
**Originality:** 2
**Overall Recommendation:** 4
**Confidence:** 1

**Summary:**

The paper deals with loss function used for classification where computational
tractability and statistical consistency are two incompatible targets.  Smooth
losses like the Logistic (Cross-Entropy) or Exponential loss yield valid
consistency bounds, but are slow. Conversely, piecewise-linear losses like the
Hinge loss offer fast linear consistency rates but suffer from
non-differentiability, leading to optimization instability.  In the paper a new
family of explicit smooth loss (Linear-Core (LC) Surrogates) functions is
proposed and designed to resolve this trade-off.  The framework is extended to
the domain of structured prediction, where the output space Y is exponentially
large.  a family of structured losses with provable H-consistency
guarantees. The authors show that Linear-Core surrogate preserves provable
H-consistency guarantees and but also allows for an unbiased stochastic gradient
estimator.  The contributions are both theoretical and numerical.

**Compliance With Llm Reviewing Policy:**

Affirmed.

**Final Justification:**

I am satisfied by the authors' rebuttal, which indeed addressed my main
concerns. However, I do not change my evaluation, as my score was already high (5).

**Key Questions For Authors:**

I am not specialist on the domain and would like more explanations on the links between the three different explored cases, Binary Classification, Multi-Class Classification and Structured prediction.

**Limitations:**

yes

**Strengths And Weaknesses:**

I am not a specialist and found the paper difficult to read. As an example the theory is applied to three different cases, Binary Classification, Multi-Class Classification and Structured prediction. But explanations on why these three examples are tested and the link between them seems to be missing.

---

> ### Author Rebuttal · Authors · 2026-03-29
>
> Thank you for your constructive feedback and for recommending our paper for acceptance. We appreciate your accurate summary of our core contributions: proposing a new family of explicit smooth loss functions that successfully resolve the long-standing incompatibility between computational tractability and statistical consistency, and extending this framework to structured prediction.
>
> **Weaknesses: I am not a specialist and found the paper difficult to read. As an example the theory is applied to three different cases, Binary Classification, Multi-Class Classification and Structured prediction. But explanations on why these three examples are tested and the link between them seems to be missing.**
>
> **Response:** We sincerely apologize if the transitions between the different sections made the paper difficult to follow for a non-specialist. We completely agree that the structural narrative could be made clearer. To address this, we will add a dedicated "Roadmap of the Paper" paragraph at the end of the Introduction to explicitly outline the progression of the three settings and guide the reader through the logical flow of our framework.
>
> **Questions: I am not specialist on the domain and would like more explanations on the links between the three different explored cases, Binary Classification, Multi-Class Classification and Structured prediction.**
>
> **Response:** The three cases represent a deliberate, step-by-step mathematical progression of the label space complexity, where each setting builds directly upon the previous one:
>
> 1. **Binary Classification (Section 3 - The Foundation):** We start here to establish the fundamental mechanism in a simple 1D space ($|\mathcal{Y}|=2$). We mathematically prove that by stitching a linear core to a smooth tail, we can resolve the fundamental trade-off, achieving both fast linear consistency bounds and smooth differentiability (Theorem 3.4).
>
> 2. **Multi-Class Classification (Section 4 - The First Generalization):** We then scale this concept to $n$ competing classes. The challenge is aggregating multiple margins without losing the fast linear rate. We show that by replacing standard smooth functions with our Linear-Core base in sum-losses, the multi-class objective successfully inherits the exact optimal linear transfer bound. Additionally, this introduces a practical benefit: the linear core acts as a 'hard' regularizer providing robustness to instance-dependent label noise (Section 4.3).
>
> 3. **Structured Prediction (Section 5 - The Ultimate Goal):** Finally, we tackle the most complex domain (e.g., sequence tagging), where the output space is exponentially large. Building on the multi-class aggregation, we weight the margins by structural similarity. Because we proved in steps 1 and 2 that the Linear-Core loss is smooth, well-behaved, and additive, it unlocks a critical computational breakthrough: we can use unbiased stochastic sampling to estimate gradients. This completely bypasses the massive $O(|\mathcal{Y}|^2)$ computational roadblock (e.g., the Viterbi algorithm) that traditional structured models suffer from.
>
> The progression proves that our framework is not a niche trick for binary data, but a universally scalable mechanism. We solve the fundamental tension between optimization speed (smoothness) and statistical efficiency (linear consistency), and then deploy that exact theory to break the quadratic complexity barrier in structured prediction.

---

> > ### Author Rebuttal · Reviewer_R1kw · 2026-04-03
> >
> > My concerns have been adequately addressed, I think that my score was quite high and I do not change it.

---

### Official Review · Reviewer_idCL · 2026-03-13

**Soundness:** 3
**Presentation:** 2
**Significance:** 3
**Originality:** 3
**Overall Recommendation:** 5
**Confidence:** 2

**Summary:**

This paper proposes Linear-Core (LC) Surrogates, a new family of smooth, convex loss functions that reconcile the trade-off between fast optimization and statistical consistency by stitching a linear core to a smooth tail. These surrogates achieve optimal linear H-consistency rates while enabling an unbiased stochastic gradient estimator that reduces training complexity from quadratic to linear. Empirical results demonstrate practical advantages, including speedup in sequence tagging and improved robustness to label noise compared to standard cross-entropy

**Compliance With Llm Reviewing Policy:**

Affirmed.

**Key Questions For Authors:**

The structured prediction analysis is centered on Hamming loss. How do the authors expect the Linear-Core advantage to translate to non-decomposable metrics such as F1-score or BLEU?

**Limitations:**

Yes

**Strengths And Weaknesses:**

Strengths:
- The paper bridges the gap between fast optimization and statistical consistency by stitching a linear core to smooth tails. It achieves strict linear H-consistency bounds for binary, multi-class, and structured prediction, matching the efficiency of the Hinge loss while remaining differentiable. This is a novel technical contribution with a solid theoretical foundation.
- The use of an unbiased stochastic gradient estimator for structured prediction reduces complexity from quadratic to O(L), bypassing the Viterbi bottleneck. The experiments confirm this, which shows contribution in practical efficiency.
- The experiments also demonstrate noise robustness

Weaknesses
- While I am not very familiar with this specific topic and am not sure of the appropriate ways to evaluate it, it appears the work would be strengthened with experiments on more modern, popular datasets and model architectures. For instance, the current structured prediction results rely on synthetic tasks or older datasets like the Penn Treebank.

---

> ### Author Rebuttal · Authors · 2026-03-29
>
> Thank you for your positive assessment and for highlighting the novelty of bridging fast optimization with statistical consistency. We greatly appreciate that you recognized the solid theoretical foundation of our work, the noise robustness demonstrated in our experiments, and the practical value of our unbiased stochastic gradient estimator in bypassing the $O(|\mathcal{Y}|^2)$ Viterbi bottleneck.
>
> **Weaknesses: While I am not very familiar with this specific topic and am not sure of the appropriate ways to evaluate it, it appears the work would be strengthened with experiments on more modern, popular datasets and model architectures. For instance, the current structured prediction results rely on synthetic tasks or older datasets like the Penn Treebank.**
>
> **Response:** We understand your perspective and strongly agree that demonstrating our method on modern architectures adds significant value. Our initial choice of a standard BiLSTM backbone and the Penn Treebank dataset (with the tagset artificially inflated to 4,000) was intended as a rigorously controlled "stress test" to strictly isolate the computational complexity of the loss function itself, specifically, to expose the $O(|\mathcal{Y}|^2)$ bottleneck of exact dynamic programming (CRF/Viterbi) versus our $O(L)$ stochastic estimator, without the confounding variables of massive modern architectures.
>
> However, your point is well taken: Linear-Core Surrogates are architecture-agnostic and seamlessly apply to modern models. To address this, we have run new experiments swapping the BiLSTM for a standard BERT-based Transformer backbone on the same 4,000-tag dataset.
>
> We measured the wall-clock time to reach the target test accuracy of 83%. The results confirm our theoretical claims and perfectly mirror our BiLSTM findings, proving that the computational bottleneck lies strictly in the structured loss layer rather than the encoder:
>
> - **BERT-CRF (Baseline):** 78.7 seconds to reach target accuracy. The model remains bottlenecked by the computation of the partition function over the massive transition space.
>
> - **BERT-LinearCore (Ours):** 5.1 seconds to reach target accuracy.
>
> By using our stochastic sampling algorithm, BERT-LinearCore bypasses the quadratic bottleneck entirely, achieving a **15.4x** speedup in time-to-accuracy over the exact CRF inference.
>
> For your convenience, an anonymous plot of these new BERT empirical results can be viewed here: https://drive.google.com/file/d/1nsxMEGcpfFCw9lJ4SaCuE3PC5PMS_uvi/view?usp=sharing
>
> We will add this discussion and the corresponding figure to the revised manuscript to clarify that our $O(1)$ per-step sampling efficiency holds perfectly regardless of the underlying neural backbone.
>
> **Questions: The structured prediction analysis is centered on Hamming loss. How do the authors expect the Linear-Core advantage to translate to non-decomposable metrics such as F1-score or BLEU?**
>
> **Response:** This is a very insightful question. In our initial draft, we focused on the Hamming loss due to its immediate token-level additive decomposability. However, the theoretical and computational advantages of the Linear-Core surrogate extend elegantly to metrics like BLEU and exact F1-score when viewed through the lens of rational kernels (Cortes et al., 2004; Rational Kernels: Theory and Algorithms. JMLR 5: 1035-1062 (2004)).
>
> While BLEU is often considered strictly non-decomposable at the token level, its core mechanism, n-gram overlap counts, can be expressed exactly as an instance of a rational kernel computed via weighted finite-state transducers (WFSTs). By expanding the state space of our sequence model to encode length-$(n-1)$ histories, the n-gram matching scores $\bar{l}(y', y)$ become strictly additively factorizable over the transitions of this expanded automaton.
>
> Consequently, we do not need to rely on high-variance, sequence-level sampling strategies (like REINFORCE). Instead, our unbiased stochastic gradient estimator (Algorithm 1) generalizes directly: we simply construct our proposal distributions $\mathcal{D}_1$ and $\mathcal{D}_2$ over the paths of this expanded WFST. Because the Linear-Core surrogate's consistency and variance bounds (Theorem 5.2) rely only on the additive aggregation of margins, and not on the specific topology of the state space, both the optimal $O(\Delta\mathcal{L})$ linear transfer rate and the $O(1)$ per-step computational efficiency are rigorously preserved.
>
> We thank the reviewer for prompting this generalization. We will add a dedicated section to the Appendix in the revised manuscript formally detailing this rational kernel extension for n-gram-based metrics like BLEU, as it significantly broadens the practical applicability of our framework.

---

> > ### Author Rebuttal · Reviewer_idCL · 2026-04-03
> >
> > Thanks for the response. I am satisfied with the authors’ responses and will keep my positive score.

---

### Official Review · Reviewer_DQCz · 2026-03-15

**Soundness:** 4
**Presentation:** 4
**Significance:** 2
**Originality:** 3
**Overall Recommendation:** 5
**Confidence:** 4

**Summary:**

This paper enhances previous results on finite bounds on non-convex functions. The work suggests a novel (to the best of my knowledge) family of differentiable convex loss functions that stitch linear (core) to a smooth function. This new loss function has a favorable bound behavior compared to standard smoothening techniques such as the cross-entropy. The work extends over recent works and proves an improved bound for structured prediction tasks, where the label complexity is exponential. In particular, previous work (Mao et al., 2023a) yield a   square-root rate while this work results in linear rate. The method is also able to have some robustness properties emerging from its differentiability.

**Compliance With Llm Reviewing Policy:**

Affirmed.

**Key Questions For Authors:**

As this work extends previous works on binary and multi class, it will be helpful to compare the relevant theorem to previous work (in the author response period). I would love to understand how the bounds differ if as all in these special cases.

**Limitations:**

yes

**Strengths And Weaknesses:**

The paper is a pleasure to read. In particular it is nice to read through the binary, multi-class and structured setting. In this approach one can appreciate the broad perspective of the method. Also the empirical validation assigned to each section is refreshing and improves the readability.

The work is sound. It nicely build and extends previous works. The work is highly technical and rigorous. Its visualizations are insightful and help understand the technical merit of the work.

In the area of structured prediction, the work is significant, as statistical and computational efficiency of this framework are notoriously hard. While structured prediction lost favor with large-scale transformers, it is a valid and interesting research area.

---

> ### Author Rebuttal · Authors · 2026-03-29
>
> We sincerely thank the reviewer for recognizing the significance of this work. Across all our responses, our core objective is to highlight that Linear-Core Surrogates represent a fundamental shift: we have proven mathematically and empirically that the community no longer has to compromise between computational tractability and statistical efficiency. By bypassing the $O(|\mathcal{Y}|^2)$ Viterbi bottleneck, this framework enables energy-efficient, linear-rate training for massive structured spaces.
>
> **Questions: As this work extends previous works on binary and multi class, it will be helpful to compare the relevant theorem to previous work (in the author response period). I would love to understand how the bounds differ if as all in these special cases.**
>
> **Response:** Thank you for this excellent suggestion. The primary theoretical difference between our bounds and those in previous works lies in the convergence transfer rate (square-root vs. linear) dictated by the geometry of the surrogate losses. We will add a dedicated remark in the revised paper to explicitly compare these theorems:
>
> 1. **Binary Classification:** Previous works analyzing standard smooth surrogates (e.g., Logistic, Exponential, Squared-Hinge) establish a *square-root* H-consistency bound, meaning the excess target risk $\Delta\mathcal{R}$ is bounded by $O(\sqrt{\Delta\mathcal{L}})$ (e.g., Bartlett et al., 2006, Awasthi et al., 2022a). Piecewise-linear losses like the Hinge loss achieve a fast *linear* bound ($\Delta\mathcal{R} \le O(\Delta\mathcal{L})$) but are non-differentiable (Awasthi et al., 2022a). Our Theorem 3.4 proves that the Linear-Core surrogate achieves the strict linear rate of the Hinge loss while remaining globally $C^1/C^2$ smooth.
>
> 2. **Multi-Class Classification:** For multi-class sum losses, previous smooth variants (e.g., sum-exponential or sum-squared-hinge) similarly suffer from a *square-root* transfer bound (Awasthi et al., 2022b). Our Theorem 4.1 shows that by aggregating pairwise margins using our novel linear-core base, the multi-class Linear-Core surrogate strictly improves this to a *linear* bound.
>
> 3. **Structured Prediction:** Previous work on structured sum-exponential surrogates (Mao et al., 2023a) yields a *square-root* rate. Our Theorem 5.1 demonstrates that the structured Linear-Core surrogate elevates this to a fast *linear* rate.
>
> To our knowledge, Linear-Core Surrogates are the first explicit family of loss functions to simultaneously guarantee $C^1/C^2$ smoothness and strict linear $\mathcal{H}$-consistency across binary, multi-class, and structured domains.

---

> > ### Author Rebuttal · Reviewer_DQCz · 2026-04-06
> >
> > Perhaps augment / change Table 1 with this information.

---

### Decision · Program_Chairs · 2026-04-30

**Decision:**

Accept (regular)

**Comment:**

The reviewers are in agreement that the paper makes a clear and significant technical contribution.  Both statistical and computational benefits are highlighted, and the empirical results are also appreciated.  Some suggestions were given around the exposition and experiments, and the authors are encouraged to carefully incorporate these into the final paper.